**Title:** Single-blind test of nine methane-sensing satellite systems from three continents
**Authors:** Evan D. Sherwin[1,a,*], Sahar H. El Abbadi[1], Philippine M. Burdeau[1], Zhan Zhang[1],
Zhenlin Chen[1],  Jeffrey S. Rutherford[1,b], Yuanlei Chen[1], Adam R. Brandt[1]
**Author Affiliations:**
[1] Department of Energy Science & Engineering, Stanford University, Stanford, California 94305,
United States
[a] Present affiliation: Lawrence Berkeley National Laboratory, Berkeley, California, 94720,
United States
[b] Present affiliation: Highwood Emissions Management, Calgary, Alberta T2P 2V1, Canada
[*] Correspondence: evansherwin@lbl.gov

## Abstract

Satellite-based remote sensing enables detection and mitigation of large point sources of climate-warming methane. These satellites will have the greatest impact if stakeholders have a clear-eyed assessment of their capabilities. We performed a single-blind test of nine methane-sensing satellites from three continents and five countries, including both commercial and government satellites. Over two months, we conducted 82 controlled methane releases during satellite overpasses. Six teams analyzed the resulting data, producing 134 estimates of methane emissions. Of these, 80 (58%) were correctly identified, with 46 true positive detections (34%) and 34 true negative non-detections (25%). There were 41 false negatives, in which teams missed a true emission, and 0 false positives, in which teams incorrectly claimed methane was present. All eight satellites that were given a nonzero emission detected methane at least once, including the first single-blind evaluation of the EnMAP, Gaofen 5, and Ziyuan 1 systems. In percent terms, quantification error across all satellites and teams is similar to aircraft-based methane remote sensing systems, with 55% of mean estimates falling within ±50% of the metered value. Although teams correctly detected emissions as low as 0.03 metric tons of methane per hour, it is unclear whether detection performance in this test is representative of real-world field performance. Full retrieval fields submitted by all teams suggest that in some cases it may be difficult to distinguish true emissions from background artifacts without a known source location. Cloud interference is significant and appears to vary across teams and satellites. This work confirms the basic efficacy of the tested satellite systems in detecting and quantifying methane, providing additional insight into detection limits and informing experimental design for future satellite-focused controlled methane release testing campaigns.

**Keywords:**
Methane, hyperspectral imaging, remote sensing, satellite, single-blind, controlled release

Satellite-based remote sensing systems continue to find large point-source emissions of climate-warming methane across the globe (Lauvaux et al., 2022; Irakulis-Loitxate et al., 2022a, b; Pandey et al., 2019; Varon et al., 2018, 2019, 2021; Sánchez-García et al., 2022). Such systems empower stakeholders in industry and government to take corrective action, both to mitigate individual sources and to inform estimates of total methane emissions, particularly in oil and natural gas systems, where many of the largest sources have been observed (Lauvaux et al., 2022; Irakulis-Loitxate et al., 2022a; Pandey et al., 2019; Varon et al., 2018; Irakulis-Loitxate et al., 2022b; Varon et al., 2021, 2019; Cusworth et al., 2022; Duren et al., 2019; Chen et al., 2022; Sherwin et al., 2023a; Sánchez-García et al., 2022).

A considerable fleet of point-source methane-sensing satellites is now in orbit, including purpose-built and repurposed instruments (Jacob et al., 2022). In the coming years, this number will increase substantially (Jacob et al., 2022).

These satellites will have the greatest positive environmental impact if their results are widely believed by a broad array of stakeholders across the world. Single-blind controlled methane release testing, in which teams estimate methane emissions from one or more metered sources without knowing the true rate, is an important and widely-used method of independently determining the capabilities of a methane sensing system (Sherwin et al., 2021, 2023b; Bell et al., 2020, 2022, 2023; Ravikumar et al., 2019; Rutherford et al., 2023).

In the first such single-blind release testing satellite systems, Sherwin et al. tested five satellites:
the commercial GHGSat-C and WorldView-3 systems and the government-supported PRISMA,
LandSat 8, and Sentinel-2 systems. In that study five teams analyzed data from different subsets
of these satellites (Sherwin et al., 2023b). This test demonstrated that, across the array of these
five satellites, this approach can be used to detect emissions ranging from 0.20 [95% confidence
interval = 0.19, 0.21] metric tons of methane per hour (henceforth t/h), for the most sensitive
systems, to 7.2 [6.8, 7.6] t/h. Relative quantification error was comparable to aircraft-based
methane sensing systems, although with significantly larger detection limits (Sherwin et al.,
2023b). Sample size was modest, however, with some satellites collecting only one
measurement, limiting generalizability of the results without additional data collection.
In addition, several methane-sensing satellites have launched since the previous test concluded in
2021, including the German EnMAP system and the 02 edition of the Chinese Gaofen 5
Advanced Hyperspectral Imager (GF5) and the 02E edition of the Ziyuan 1 Advanced
Hyperspectral Imager (ZY1) (EnMAP, 2023; Xinhua, 2022; Song et al., 2022). Although these
satellites were not primarily designed to sense methane, scientists have used similar systems to
detect substantial methane point sources from oil and natural gas infrastructure (Irakulis-Loitxate
et al., 2021).
This work conducts single-blind testing of nine distinct satellite systems, focusing on detection
and quantification performance for releases ranging from 0.03-1.6 t/h. In addition, we take steps
to evaluate the generalizability of our results through a highly transparent experimental design, in
which all teams submit full methane retrievals for the scene surrounding the release. This
approach provides insight into which identified methane emissions are unambiguously detected
and which might be difficult to distinguish from artifacts if the source location were not known.
**1      Materials and Methods**
We employed a fixed-location single-blind controlled methane release experimental design to
evaluate point-source methane sensing systems from October 10th to November 30th, 2022.
Participating teams were aware of the existence, timeframe, and precise location coordinates of
the test site. Teams were not informed during a given observation whether gas would be
released, nor of the size of released emissions. Teams were informed of an approximate upper
bound of 1.5 t/h. Teams were not given the precise configuration of Stanford equipment on the
ground, though large equipment may have been visible from space in some cases.
Metered controlled release volumes – including releases with zero volume – were retained by the
Stanford team and not given to teams until all estimates were submitted by all participants for all
stages of the test. Analysts estimated the presence and magnitude of methane emissions for each
overpass, with a reporting approach in compliance with the Advancing Development of
Emissions Detection (ADED) protocol for airplane and satellite systems (Zimmerle, 2022). More
information is provided in the Supporting Information (SI), Section S1.1.
We performed releases during overpasses of nine satellite constellations: the commercial
satellites GHGSat-C (GSC) of Canada and the US-based WorldView-3 (WV3), as well as
publicly-funded satellites, including the German Environmental Mapping and Analysis Program
(EnMAP), the Chinese Gaofen 5 (GF5), Ziyuan 1 (ZY1), and Huanjing 2 (HJ2), the Italian
PRecursore IperSpettrale della Missione Applicativa (PRISMA), the US LandSat (LS) 8 and 9,
and the pan-European Sentinel-2 (ESA, 2022a, b, c; Jervis et al., 2021; OHBI, 2022; EnMAP,
2023; Liu et al., 2019; USGS, 2022; Song et al., 2022; Zhong et al., 2021). With the exception of
the GHGSat-C constellation, none of these satellites was explicitly designed for methane
sensing, but their data have instead been applied to this end. Analysis teams first attempted to
estimate emissions volumes using available data from satellites and wind reanalysis products. In
some cases, multiple teams assessed the same observation from an instrument, providing an
opportunity to empirically assess variability due to source quantification algorithms, which
participating teams were not required to release. See the SI, Section S3 for the details each team
elected to share about their algorithms.
These satellites range from high-sensitivity/narrow swath to low-sensitivity/large swath, as
illustrated in Table 1. Revisit time is also anticorrelated with instrument sensitivity. The
Sentinel-2 and LandSat 8/9 systems have estimated detection limits of roughly 1-5 t/h (Gorroño
et al., 2023), but each satellite in these constellations covers the bulk of the world's landmass
every 10-16 days with a swath of 185-290 km (USGS, 2022; ESA, 2021a). GHGSat, EnMAP,
GF5, PRISMA, WorldView-3, and ZY1 are targeted "point-and-shoot" systems, with higher
resolution but narrower swaths of 12-60 km (ESA, 2022a, b; Jervis et al., 2021; OHBI, 2022;
EnMAP, 2023; Liu et al., 2019; Song et al., 2022). Existing publicly available information does
not specify whether HJ2 is targeted or has global coverage, but its swath of 800 km suggests it is
capable of global coverage (Zhong et al., 2021). Pixel size also varies widely across satellites,
with most tested satellites ranging from 20-30 m square pixels, while HJ2 has 6 km square pixels
and WorldView-3 has highly sensitive 3.7 m square pixels. Spectral resolution varies as well
across the tested satellites, from 0.3 nm for GHGSat-C and 200 nm for Sentinel-2 and LandSat
8/9 (Jacob et al., 2022), discussed further in the SI, Section S2. See the SI, Section S2 for
additional discussion of the capabilities of each satellite system.
Table 1. Key characteristics of each participating satellite constellation, from lowest to highest swath width, which is roughly proportional to an instrument's
minimum methane detection limit. Global coverage refers to a configuration that passively covers most of Earth's surface over some number of orbits, while
targeted coverage refers to a "point-and-shoot" instrument that must be pointed to a particular location. Nadir pixel size is presented here. Constellation size
includes only active satellites. Accessing data from the GF5, ZY1, and HJ2 satellites requires permission from the Chinese government. Adapted with permission
from (Sherwin et al., 2023b).

| Satellite | Coverage | Constellation size | Swath [km] | Pixel size [m] | ~Revisit time (per satellite) | Data availability | Source |
|---|---|---|---|---|---|---|---|
| GHGSat-C | Targeted | 8[§] | 12 | 25x25 | 14 days | Commercial | (ESA, 2022a; Jervis et al., 2021) |
| WorldView-3 | Targeted | 1 | 13.1 | 3.7x3.7 | 1 day[‡] | Commercial | (ESA, 2022b) |
| PRISMA | Targeted | 1 | 30 | 30x30 | 7 days | Public | (OHBI, 2022; ESA, 2012) |
| EnMAP | Targeted | 1 | 30 | 30x30 | 4 days[†] | Public | (EnMAP, 2023) |
| Gaofen 5 (GF5) | Targeted | 1 | 60 | 30x30 | 5-8 days[*] | Government | (Liu et al., 2019; Zhang et al., 2022; Luo et al., 2023) |
| Ziyuan 1 (ZY1) | Targeted | 1 | 60 | 30x30 | 1-3 days[*] | Government | (Song et al., 2022) |
| Landsat 8/9 | Global | 2 | 185 | 30x30 | 16 days | Public | (USGS, 2022) |
| Sentinel-2 | Global | 2 | 290 | 20x20 | 10 days | Public | (ESA, 2021a) |
| Huanjing 2 (HJ2) | Unknown | 2 | 800 | 6x6 km | ≤4 days[*] | Government | (Zhong et al., 2021) |

[§]Three of these GHGSat C satellites were launched after the conclusion of testing.
[‡]WorldView-3 requires a 4.5-day repetition cycle for best resolution within 20° off nadir.
[†]EnMAP requires a 27-day repetition cycle for best resolution within 30° off (Jacob et al., 2022).
[*]Revisit times for GF5, ZY1, and HJ2 are inferred, at least in part, from overpass schedules submitted by NJU.
This test does not include the TROPOMI system on the Sentinel-5P satellite, which has a
detection limit far above the maximum of the release apparatus used in this study (ESA, 2021b).
We inquired about tasking the Earth Surface Mineral Dust Source Investigation (EMIT) satellite,
launched by the US National Aeronautics and Space Administration (NASA) in July 2022
(Wang and Lee, 2022), but the system was not available to participate in this test.
Participating analysis teams include private companies GHGSat (GHGSat, 2022), Kayrros
(Kayrros, 2022), Maxar (Scott, 2022), and Orbio Earth (Orbio, 2023), as well as the Land and
Atmosphere Remote Sensing (LARS) group of university researchers from Universitat
Politècnica de València [Luís Guanter, Javier Roger Juan, and Javier Gorroño Viñegla (Irakulis-
Loitxate et al., 2022a, b, 2021; Guanter et al., 2021)] and Nanjing University [Fei Li, Huilin
Chen, and Yongguang Zhang (Jia et al., 2022)]. Each analysis team had the opportunity to
submit estimates for all satellites tested, with the exception of the GHGSat-C satellites, to which
GHGSat had sole access. See the SI, Section S3 for a description of each team and its members,
as well as a list of invited teams that declined to participate.

## 1.1    Materials

For the full test period, our experimental equipment was located near Casa Grande, Arizona, south of Phoenix, Arizona in the United States, with the release stacks located at [32.8218205°, -111.7857730°].

The methane source was two trailers of compressed natural gas, shown in Figure 1, which passed through a pressure regulation and reheating apparatus. The gas was then transmitted to the metering and release trailer via a 7.62 cm (3 in) shipping hose at an exit pressure of roughly 150-200 psig (1.03-1.37 Mpa), passing through one of three possible Coriolis meters before release through one of two stacks, at a release height of either 7.3 or 3.0 m above ground level (El Abbadi et al., 2023), shown in the SI, Figure 1. This testing setup approximately mimics an unlit flare or tank vent on an oil and gas production site or other facility.

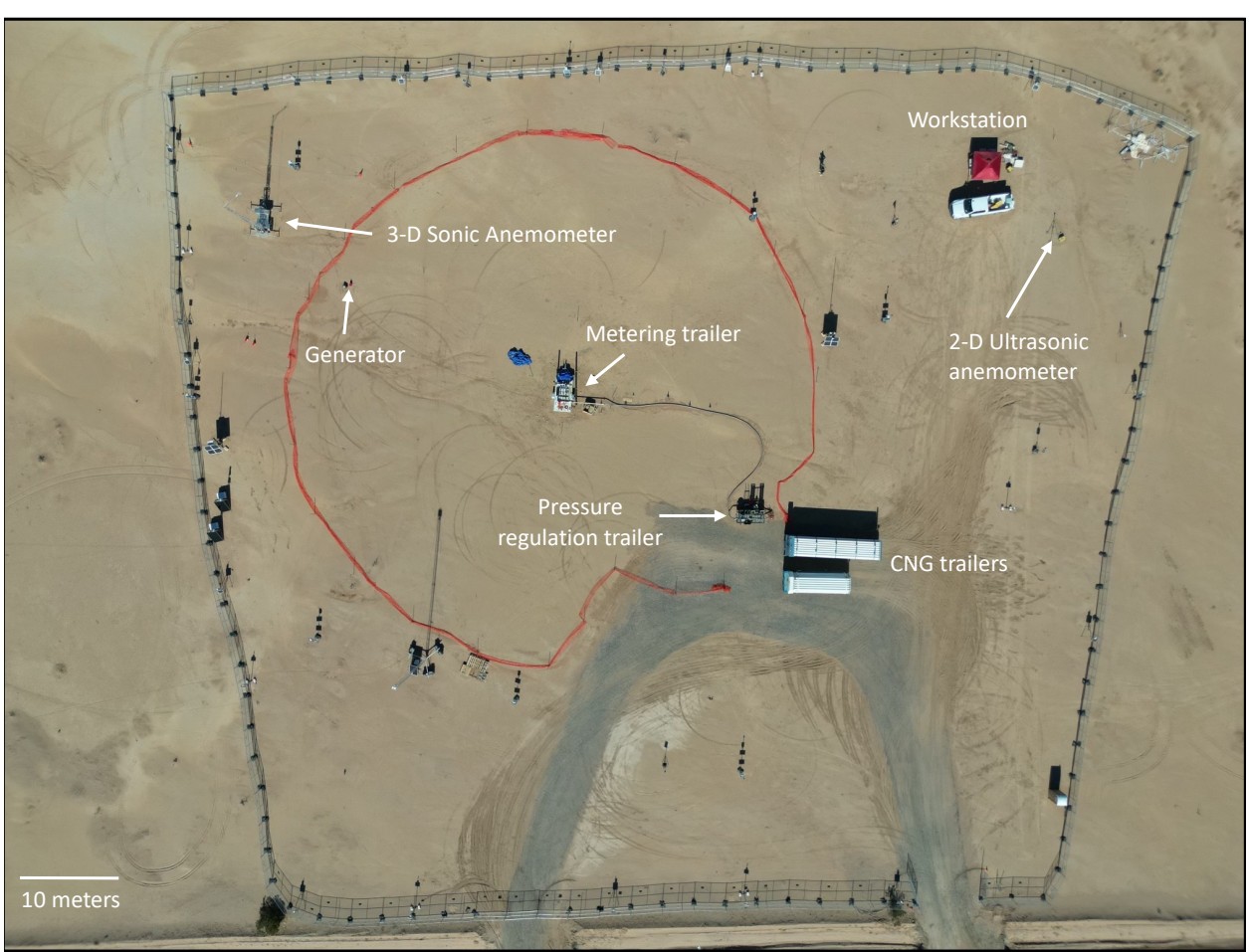

Figure 1. Aerial photograph of the site. Note that the workstation is ~60 m from the release apparatus and ~50m from the compressed natural gas (CNG) trailers. Reproduced with permission from (El Abbadi et al., 2023).

This experiment was designed to provide near-optimal conditions for methane-sensing satellites. In addition to the desert background, the site contained only equipment necessary to conduct controlled methane releases and test a suite of methane sensing technologies. The result is a

significantly less complex scene than many oil and gas facilities, which will often contain
multiple pieces of infrastructure such as wellheads, tanks, flares, and separators at production
sites, and entire buildings with sophisticated machinery and piping at compressor stations and
gas processing plants. More complex scenery can make methane remote sensing more
challenging. Future work with scenes that more closely mimic industrial sites will help determine
the associated differences in technology efficacy, if any.
Achievable release rates for the three Coriolis meters, installed in pipes of different diameter,
were 2 – 30 kilograms per hour (kg/h), 30 – 300 kg/h, and 300 – 2,000 kg/h for natural gas. See
(El Abbadi et al., 2023) for further detail.

## 1.2    Safety

All natural gas equipment fabrication, operation, and transportation was conducted by personnel
affiliated with Rawhide Leasing, a gas services contractor. Stanford personnel contributed to
assembly of some equipment, but did not operate natural gas release equipment or pass within
our 100-foot (30.5 m) safety perimeter fence during active releases. The research workstation,
from which Stanford researchers coordinated data collection and related field operations, was
~60 m away from any equipment through which natural gas flowed.
In addition, Stanford researchers periodically monitored plume dissipation in real time via a
FLIR GasFinder 320 infrared camera and continuously paid attention to olfactory signals from
the gas, which was odorized. The infrared camera showed clearly that the plume dissipated well
before reaching any on-site personnel. Equipment design contributed to this intrinsic safety,
because the emission source was elevated off the ground and gas often exited at a high vertical
velocity, particularly at larger release volumes, accelerating natural methane lofting. When
Stanford researchers detected gas smell during testing, they diligently checked infrared footage
of the plume and/or ambient wind conditions to ensure safety of all personnel onsite.

## 1.3    Data logging

Stanford researchers collected data logs directly from the Coriolis gas flow meters, accounting
for modest timestamp offsets as described in (El Abbadi et al., 2023).

## 1.4    Data collection procedures

All satellite-coincident releases began at least 15 minutes before the scheduled satellite overpass
time, provided by participating teams.

Stanford personnel set all release levels remotely, using WiFi-enabled control software deployed
on a laptop computer. For releases conducted on or before October 20[th], Stanford personnel set a
desired flow rate, with an automated control system adjusting valves in real-time to target that
rate. After it became clear that this approach resulted in unnecessary flow rate variability,
releases from October 21[st] on were conducted by setting the relevant valve to a desired level of
openness, improving flow stability while slightly reducing the system's ability to target a specific
release rate, although this system still represents a major improvement over the manual approach
employed in (Sherwin et al., 2023b). Flow can fluctuate during the releases due to shifts in
pressure, temperature, and simple turbulent flow through the system. All performed releases
except four had flow variability with a 5-minute 95% confidence interval within ±10% of mean
flow. On November 15th, a GF5 satellite acquisition was rescheduled without notice to the
Stanford team for a time that happened to be one minute after conclusion of a different satellite
release, resulting in flow variability within ±20% of the 5-minute mean. Three additional
releases exceeded a 5-minute flow variability 95% confidence interval of ±10%: the October 11th
GHGSat-C overpass (in which the instrument was not tasked), the October 17th WorldView-3
release of 0.042 [0.034, 0.050] t/h, and the November 30th PRISMA release of 0.98 [0.87, 1.08]
t/h.
Interference from other sources was examined and found to be minimal. Over the course of the
experiment, we tested the Carbon Mapper, GHGSat AV, Kairos Aerospace, MethaneAIR, and
Scientific Aviation aerial methane sensing systems (El Abbadi et al., 2023), all of which are
more sensitive than any of the satellites tested. These aircraft, which also surveyed the nearby
area during the process of data collection, found no detectable methane sources outside our test
site. This strongly suggests that our test was free of interference from significant confounding
methane sources. The only evidence of modest possible landfill interference comes Scientific
Aviation, whose highly sensitive in situ measurement technology found modest and diffuse
methane concentration enhancements over a nearby landfill, potentially impacting only one of
the three days of testing, and only one of the seven measurements conducted on that day (El
Abbadi et al., 2023).

## 1.5    Flow rate uncertainty

Sources of uncertainty in measured methane flow rates include variability in actual natural gas
flow rates (represented as the standard deviation of metered natural gas flow over a 5-minute
period), rated meter uncertainty, and uncertainty in gas composition, which can vary even for a
consistent supplier. We used highly precise Coriolis meters, which have manufacturer rated
uncertainty of 0.25% of the flow rates used in this study (El Abbadi et al., 2023). Natural gas
composition for the gas used in these releases, derived from measurement stations on the
transmission pipeline that supplied the gas used in this test, ranged between 93.6% [93.3%,
93.9%] and 95.4% [94.7%, 96.1%] methane, described further in the SI, Section S1.2 and in
reference (El Abbadi et al., 2023). We propagate these sources of error into our metered values
using code listed in data and code availability statement. See (El Abbadi et al., 2023) for further
discussion of sources of metering uncertainty and our method of determining flow rate
uncertainty, as well as detailed gas composition data.
Following (Sherwin et al., 2023b), we use a 5-minute averaging period used to compute flow
variability. This is based on the fact that a plume traveling with a relatively slow average wind
speed of 2 m/s, the minimum observed 5-minute average wind speed for any valid satellite
measurement, would traverse 600 m within 5 minutes (300 seconds). By this distance, much of
the originally emitted methane has likely dissipated into background concentrations, with the
bulk of the methane enhancement detected by a satellite remaining closer to the release point.

## 1.6    Experimental design

This single-blind field trial employed a two-stage experimental design, modeled on (Sherwin et
al., 2023b). This approach aims to disentangle the effect of wind speed uncertainty from other
sources of methane quantification uncertainty, e.g. due to algorithmic differences.

Stanford personnel released metered quantities of methane from the test site via procedures
described above and in reference (El Abbadi et al., 2023). The Stanford ground team and
contract personnel operating equipment communicated no information to participating teams
regarding metered flow rates or metered wind speed or direction. Participating teams were aware
of the precise location coordinates of the test, but were not informed of the precise configuration
of ground-based equipment within the test site. Teams were given a rough range of possible
overall flow rates, from below 0.01 t/h to roughly 1.5 t/h. To facilitate efficient tasking of
government satellites, LARS and NJU were informed in advance that weekend releases in
November would be cancelled and all such dates were excluded from single-blind analysis for
those teams. In addition, participating teams were not informed of the details of the equipment or
its configuration, or the diameter of the pipes and hoses involved, although teams were informed
that the test would use compressed natural gas as the methane source.

After each team submitted final stage 1 estimates based on the above information, we proceeded
to stage 2 estimates. In stage 2, Stanford provided 10 m wind speed and direction data from our
on-site ultrasonic anemometer (shown in Figure 1) at one-second resolution and teams were
allowed to re-estimate emissions based on measured ground wind conditions rather than re-
analysis products as in stage 1. All teams submitted stage 1 and stage 2 estimates, with the
respective timelines described in the SI, Section S2.10. Note that turnaround time for results in
this study may not be representative of commercial or field performance.

## 2 All tested satellites detected methane

For the eight satellites given nonzero methane emissions, at least one analysis team correctly detected methane. The single HJ2 measurement, using the HJ2B satellite, was rescheduled without notice to a time in which Stanford was not releasing methane.

In total, the nine tested satellites conducted 82 overpasses. Six analysis teams analyzed data from between 1 – 8 satellites each, resulting in a total of 492 potential estimates. Stanford filtered many of these estimates from analysis before teams submitted results, for various reasons (e.g. due to release system malfunction or prior notice to teams tasking government satellites that there would be no weekend releases in November). In addition, most teams opted to submit estimates for only a subset of all available satellites. See the SI, Section S1.3 for further discussion of data exclusion criteria.

Of the 139 estimates not filtered by Stanford, in five instances (3.6% of the total), teams filtered estimates using internal quality control criteria related to cloud cover, image clipping, or other factors that could compromise the ability to produce a valid methane estimate. GHGSat filtered three retrievals from the GHGSat-C satellite due to clouds (see Figure 8 and the SI, Section S4 for sky images and further discussion of clouds). LARS filtered two WorldView-3 retrievals due to cloud cover (November 22[th]) and inconsistent wind, and possible effects of human-made surface features (October 10[th]). As a result, a total of 134 estimates included valid methane detection estimates.

Of these 134 estimates, 80 (58%) were identified as either a true positive or true negative, correctly determining the presence or absence of methane, as shown in Figure 2. True positives represent 46 (34%) of total estimates with valid detection estimates, with 34 (25%) true negatives. Note that for Sentinel-2, we consider non-detection of an 0.005 t/hr release on November 28[th] to be a true negative, as this value is more than two orders of magnitude below existing estimates of the detection threshold of this system (Gorroño et al., 2023; Sherwin et al., 2023b).

185

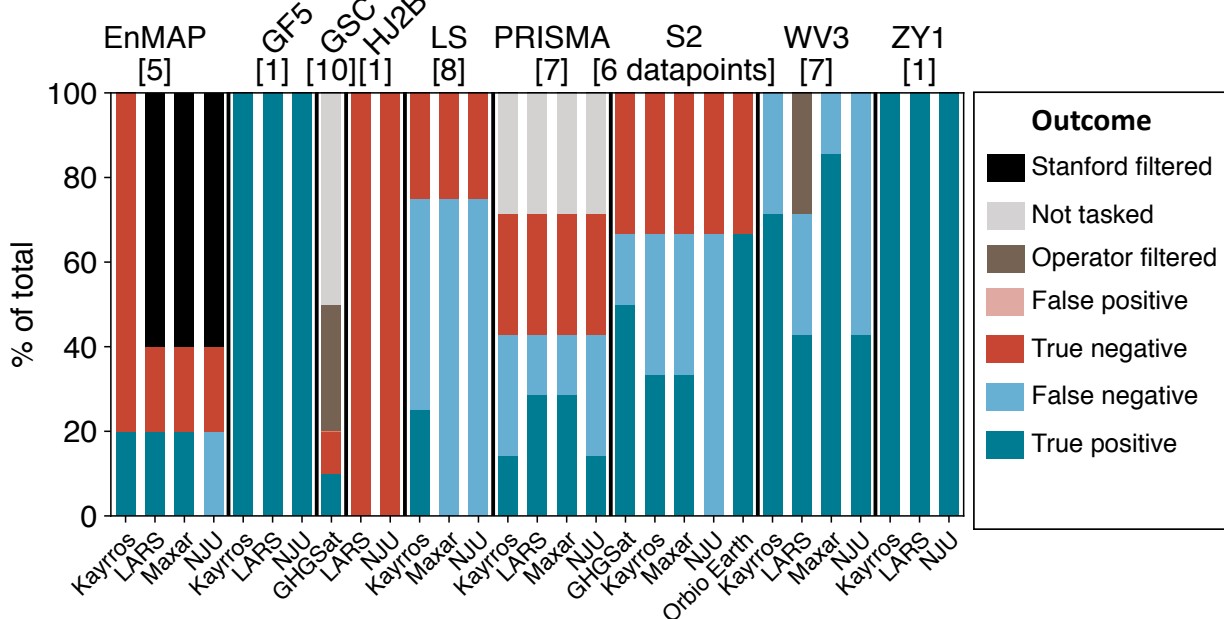

Figure 2. Detection performance by satellite and team. The total number of measurements per satellite is listed in
brackets, excluding measurements filtered by Stanford across all teams. All teams analyzing data from the three
Chinese satellites, Gaofen 5 Advanced Hyperspectral Imager (GF5), Ziyuan 1 (ZY1), and Huanjing 2B (HJ2B) all
correctly classified all emissions. Detection performance varied substantially across the Sentinel-2 (S2) and LandSat
8/9 (LS) wide-area satellites. On several days, anticipated measurements from PRISMA and GHGSat-C (GSC) were
not collected because the satellite was not tasked. In others, e.g. two WorldView-3 retrievals from LARS, no
retrieval was conducted due to concerns over image clipping or excessive cloud cover. No teams submitted false
positives, in which they reported the presence of methane when none was released.

Of the 41 false negatives (30%), most (25) are concentrated in the lower-sensitivity Sentinel-2
and LandSat 8/9 systems. There is substantial variability in false negative rates across teams. For
example, Orbio Earth correctly classifying all valid Sentinel-2 releases. GHGSat missed only one
Sentinel-2 release, and NJU detected none. This highlights that analysis of identical spectral data
can produce very different results. As in (Sherwin et al., 2023b), there were no false positives,
defined as incorrect reports of the presence of methane.

In several cases, a satellite was not tasked during an overpass for which the Stanford team
conducted a release, either due to technical issues, scheduling issues, or miscommunications
between the Stanford team and the operator. This occurred for five GHGSat overpasses and two
PRISMA overpasses, resulting in a total of 13 Not Tasked estimates from participating teams for
these two satellites, 9% of all estimates not filtered by Stanford.

## 2.1 First-time single-blind detections from three satellites

This work includes the first-ever single-blind test of the Chinese Ziyuan 1 (ZY1), Gaofen 5
(GF5), and Huanjing 2B satellites (HJ2B), as well as the European EnMAP satellite. Previous
studies have used a subset of these satellites to detect and quantify point-source emissions with
estimated magnitudes as small as 0.5 t/h, but have not performed ground-truth testing (Irakulis-
Loitxate et al., 2021). Roger et al. compare EnMAP retrievals with the single-blind-validated

PRISMA satellite as a benchmark, finding promising results, especially for offshore emissions of
1 t/h or more (Roger et al., 2023).
Figure 3 shows masked methane plume images from ZY1, GF5, and EnMAP, over a standard
optical satellite image background, for emissions of roughly 1 t/h. Masking refers to the process
of spatially differentiating a methane emission from background noise. The HJ2B acquisition
was rescheduled without prior notice to the Stanford team to a time at which no release took
place, which all teams analyzing HJ2B data correctly identified as a non-emission. We present
images from all teams analyzing satellite data from these measurements, including LARS,
Kayrros, NJU, and Maxar. See the SI, Section S4 for masked and unmasked plume images for all
satellites and teams.

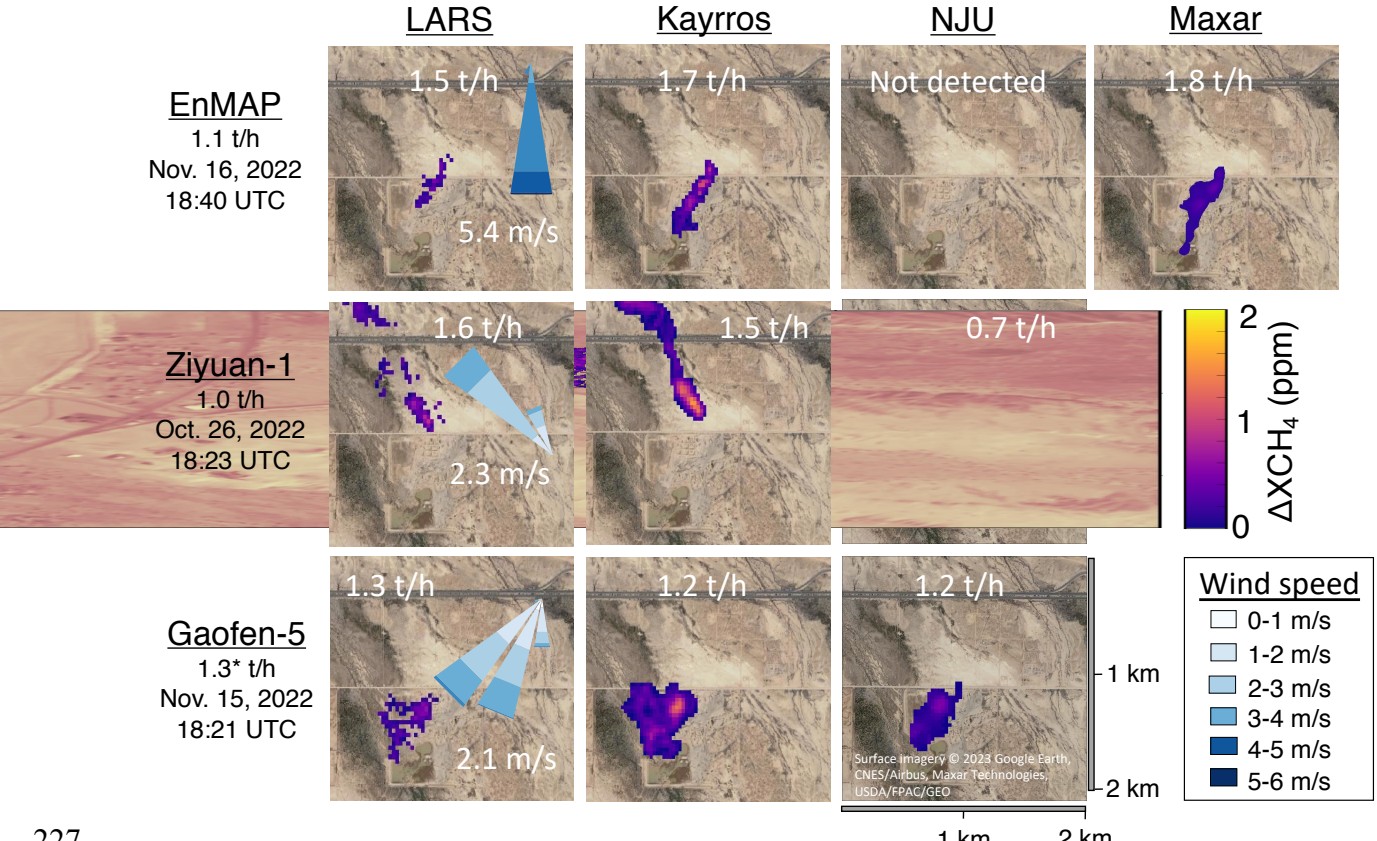

Figure 3. Visualization of detected emissions for the newly-tested European and Chinese satellites, using the release
closest to 1 t/h in all cases. The true measured emission rate, as well as the timestamp are shown below the satellite
name. Mean estimated volume from each team/satellite pair, as well as a 5-minute wind rose of measured 10-meter
wind speed and the direction toward which the wind was blowing, are superimposed on the corresponding picture.
The wind rose represents a histogram of one-second wind measurements in each direction, broken down by wind
speed. Where an emission was not detected, we show the full unmasked retrieval field. Cloud-free surface imagery
© 2023 Google Earth, CNES/Airbus, Maxar Technologies, USDA/FPAC/GEO. *The Gaofen 5 measurement was
rescheduled without notice to a time that happened to be one minute after releases had concluded for a different
satellite, resulting in artificially high variability in the metered ground-truth flow rate.
Note that, as was observed in (Sherwin et al., 2023b), teams analyzing precisely the same
spectral data can produce methane plume masks with very different shapes. Each row represents
a distinct satellite, while each column shows estimates from a distinct team. For example, the
first row shows estimates for the November 16th EnMAP satellite measurement, for which four
teams submitted estimates. Three of the four teams detected the emission. LARS, Kayrros, and
Maxar all show masked plumes traveling in roughly the same direction, but the Kayrros and
Maxar plumes are fairly contiguous, while the LARS plume is smaller and contains disjunct or
tenuously-connected clusters of estimated methane enhancements. Overall, masks from LARS
are more conservative and less spatially contiguous than other teams. However, quantification
estimates from LARS, Kayrros, and Maxar all have overlapping quantification intervals,
demonstrating that the results are not statistically distinguishable across these three teams (NJU
did not detect this EnMAP emission). Even with cases with large mean differences, e.g. October
26th estimates for ZY1, which range from 1.6 [1.2, 2.0] t/h for LARS to 0.7 [0.6, 0.9] t/h for
NJU, the 95% confidence intervals overlap. These findings suggest that many factors influence
quantification performance, even when working with identical spectral data, but large
uncertainties make disentangling these differences a challenge. Further analysis of these
algorithmic differences is beyond the scope of this work, as teams were not asked to provide
algorithmic details, which are often proprietary. Further experimentation may enable analysis of
general trends in advantages of one algorithm over another, but the order-100 number of
datapoints here is insufficient to make such judgements.
Wind can vary substantially in speed and direction even on five-minute timescales relevant to
methane quantification, as shown in wind roses inset in the left-most panel for each satellite in
Figure 3. This variability clearly influences plume formation, with emissions with steadier wind
directions and higher speed, such as the EnMAP and ZY1 measurements shown here (5.4 [3.7,
7.2] m/s and 2.3 [1.0, 3.7] m/s average wind speed, with a wind direction circular standard
deviation of 16° and 11°, respectively), resulting in narrower plumes. The highlighted GF5
measurement has slower and more variable winds and a wider plume in all three retrievals (2.1
[0.3, 4.0] m/s, with a wind direction circular standard deviation of 18°).
## 2.2    Reliable overall quantification performance
Releases in this study covered a wide range of emission rates, as low as 0.0332 [0.0328, 0.0336]
t/h, analogous to a medium-sized liquids unloading event at an oil and gas production site (Bell
et al., 2017), and as high as 1.48 [1.43, 1.52] t/h, analogous to a medium-sized unlit flare
(Cusworth et al., 2021). For all detected emissions, mean estimates for all satellite-team
combinations are between -56% and 456% of the metered value (Figure 4; see also SI, Section
S5), with 55% of nonzero estimates falling within ±50% of the metered value. Excluding
estimates from Maxar, which discovered after submitting results that its estimates were likely a
factor of 2.3 too high due to a misinterpretation of a deprecated spectral absorption library, this
fraction rises to 63% (Hayden and Christy, 2023). However, the best-fit line across all satellite
measurements, any one of which may have substantial quantification error, is largely unbiased,
with a slope close to the ideal value of 1 (which would denote perfect agreement on average).

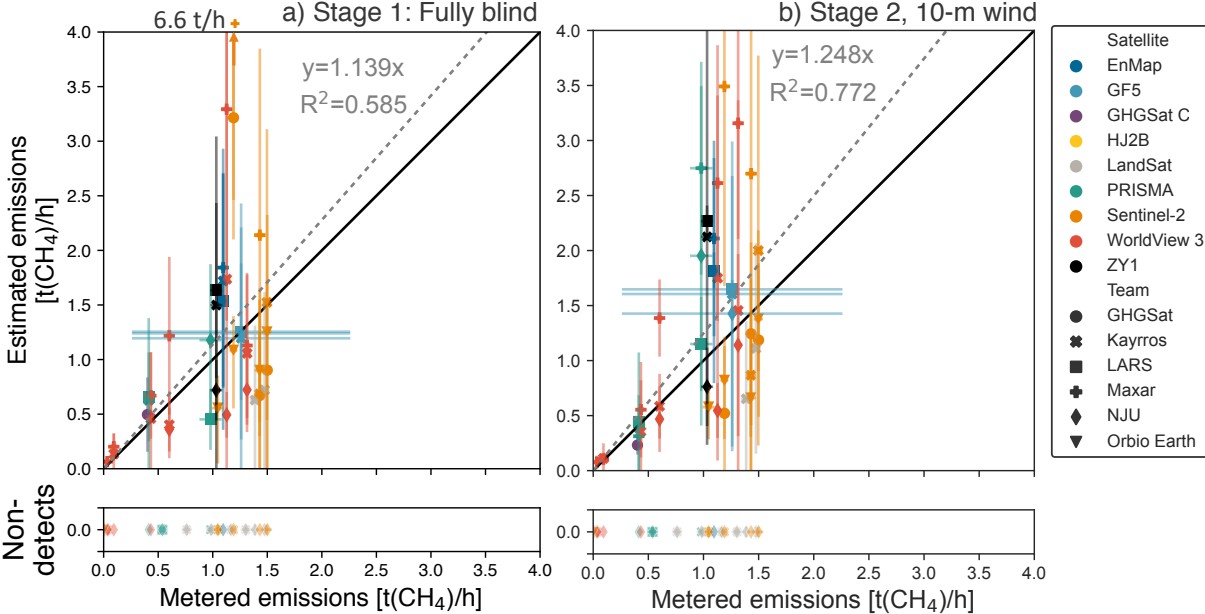

Figure 4. Methane quantification performance by satellite and team. Metered emissions compared with single-blind
estimates for each overpass with successfully reported data, with 95% X and Y confidence intervals. a) Fully blind
stage 1 results using modeled wind speed estimates. Note one Sentinel-2 estimate exceeds the y-axis limit at 6.6
t($CH_4$)/h. b) Stage 2 results using on-site 10 m wind speed and direction measurements. LARS WorldView-3
quantification estimates are excluded from the main analysis, as stage 1 estimates were submitted after wind data
had been unblinded to a member of the LARS team not involved in analyzing WorldView-3 data, while
corresponding stage 2 estimates were submitted after release volumes were unblinded. Note that Maxar submitted
PRISMA estimates for stage 2 only. The grey dashed lines represent an ordinary least squares fit with the intercept
fixed at zero, with slope and uncentered $R^2$ displayed. Maxar has since determined that its estimates were likely
artificially high, potentially introducing upward bias into aggregate statistics (Hayden and Christy, 2023). See the SI,
Section S4.2 for a version of this plot excluding Maxar, which shows overall improvement in both slope and $R^2$. The
black solid lines denote exact 1:1 agreement. See the SI, Section S4 for satellite- and team-specific results.
In percent quantification error terms, this overall performance approaches that of the satellites
and teams tested in Sherwin et al. 2023, in which 75% of estimates fell within ±50% of the
metered value, demonstrating a relative error profile similar to that observed in aircraft-based
methane remote sensing technologies (albeit with minimum detection limits one to three orders
of magnitude larger) (Sherwin et al., 2023b; El Abbadi et al., 2023; Bell et al., 2022). Direct
comparison with the results in Sherwin et al. 2023 is complicated by the fact that releases in this
study focused on smaller emissions, with a maximum of roughly 1.5 t/h instead of 7.2 t/h.
Aircraft-based methane remote sensing technologies tested in El Abbadi et al. tend to have
modestly better quantification performance in percentage terms, with 68-80% of estimates from
Carbon Mapper, GHGSat, Kairos Aerospace, and MethaneAIR falling within ±50% of the
metered value (El Abbadi et al., 2023), a substantial improvement over prior tests of the same
technologies (Sherwin et al., 2021; Rutherford et al., 2023). In each of these cases, best-fit lines
have a slope that is similarly close to 1:1 agreement.
See the SI, Section S4 for error summary statistics by satellite and team. Error bars in metered
values along the x-axis are generally too small to be visible, with the notable exception of the
GF5 measurement, which was rescheduled without notice to a time that happened to be one
minute after releases had concluded for a different satellite.
In stage 2 of the test, teams produced updated results using measured 10 m wind data from an
on-site three-dimensional ultrasonic anemometer, though still blind to released volumes.
Applying an ordinary least squares linear fit to all quantified emissions, with the intercept set to
zero, we see a modest increase in slope, rising from 1.139 [0.832, 1.446] in stage 1 to 1.248
[1.037, 1.459] in stage 2 (Figure 4).
Interpretation of these results is complicated by the fact that the Maxar team discovered after
submitting blinded results that the spectral library underlying their estimates contained an error
that likely artificially inflated their estimates by a factor of 2.3, discussed in detail in a white
paper produced by Maxar personnel (Hayden and Christy, 2023). This is consistent with the
Maxar-specific parity chart in the SI, Section S4, alongside other satellite- and team-specific
results, which shows a regression best fit line of 2.334 [1.030, 3.638] and an uncentered $R^2$ of
0.96, indicating a close linear fit. Excluding Maxar results (as in the SI, Section S4.2), the Stage
1 slope for all remaining teams falls to 0.897 [0.716, 1.078], with a Stage 2 slope of 1.010
[0.841, 1.180], almost perfect average agreement with metered values. These slopes are 21% and
19% below the respective estimates in which Maxar values were included.
Note that LARS WorldView-3 quantification estimates are excluded from the main analysis, as
stage 1 estimates were submitted after wind data had been unblinded to a member of the LARS
team not involved in analyzing WorldView-3 data, while corresponding stage 2 estimates were
submitted after release volumes were unblinded. Although the Stanford team believes all LARS
quantification estimates for WorldView-3 were submitted without leveraging unblinded data, we
must exclude them from the main analysis. This does not affect the integrity of detection
estimates, as only wind measurements were unblinded when these were first submitted. See the
SI, Section S4 for LARS WorldView-3 quantification results.
After incorporating on-site wind measurements, the uncentered $R^2$ increases from 0.585 to 0.772,
a substantial improvement in goodness-of-fit. Excluding Maxar results, these numbers rise to
0.768 and 0.826, respectively. The linear fit presented here treats all estimated emission rates
from all team as independent datapoints. Note that uncentered $R^2$ values from such a linear fit,
with a zero intercept, have a different interpretation than $R^2$ values from nonzero-intercept
regressions and should not be compared directly. See (Sherwin et al., 2023b), SI Section S5 for
further explanation of the reasons for an ordinary least squares fit with the intercept fixed to zero.
This improved average linear fit with in situ wind does not necessarily translate to lower error for
each individual satellite, as shown in the SI, Section S4, alongside additional regression results.
Confidence intervals submitted by teams appear to be modestly overconfident. For Stage 1
estimates, the metered value is within the provided 95% confidence interval only 70% of the
time, somewhat below the expected value of 95% for perfectly-calibrated 95% confidence
intervals. For Stage 2, this fraction falls to 52%, although mean error improves. Note that these
values combine results from multiple satellites and teams, and thus represent an overall sense of
the performance of satellite-based methane sensing systems as a technology class. Additional
data collection is needed to characterize the performance of each individual satellite in detail.
Figure 5 shows Stage 1 fully blinded results, the same underlying data as in Figure 4, for each
individual team. Team-specific parity lines tend to fall near the ideal 1:1 level, with Orbio Earth
and NJU exhibiting modest low bias parity slopes of 0.74. Note that Maxar's parity slope of 2.3
matches almost exactly with the factor of 2.3 they believe was error introduced into their system
through misinterpretation of a deprecated spectral library (Hayden and Christy, 2023). The bulk
of false negatives were from the relatively low-resolution Sentinel-2 and LandSat 8/9 satellites.
However, Orbio Earth successfully detected all Sentinel-2 releases, except a release below 0.010
t/h (testing another technology), far below all estimates of the Sentinel-2 detection limit
(Gorroño et al., 2023; Sherwin et al., 2023b). These results highlight algorithmic variation across
teams analyzing the same spectral data.

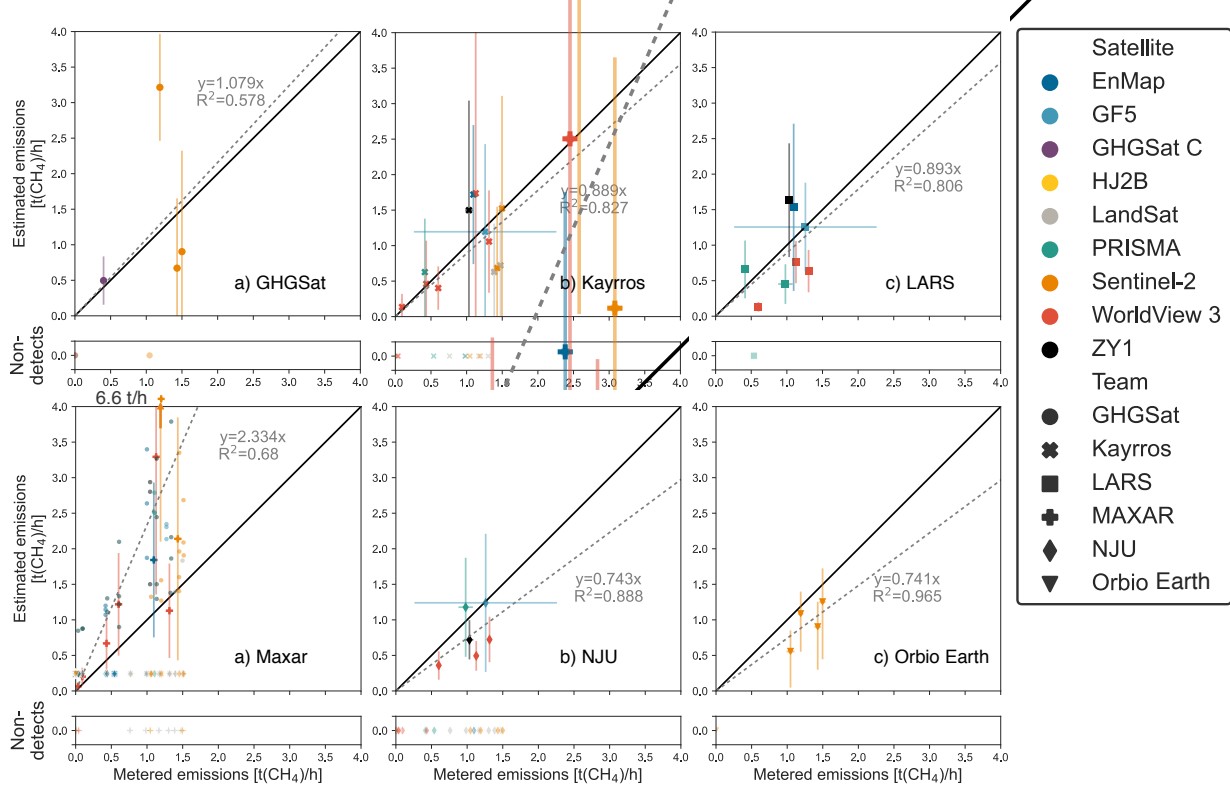

Figure 5. Parity charts by team, for fully blinded Stage 1 estimates only. Metered emissions compared with single-blind estimates for each overpass with successfully reported data, with 95% X and Y confidence intervals. Note one Maxar Sentinel-2 estimate exceeds the y-axis limit at 6.6 t(CH₄)/h. LARS stage 1 WorldView-3 quantification estimates are excluded from the main analysis, as they were submitted after wind data had been unblinded to a member of the LARS team not involved in analyzing WorldView-3 data. The grey dashed lines represent an ordinary least squares fit with the intercept fixed at zero, with slope and uncentered $R^2$ displayed. Maxar has since determined that its estimates were likely artificially high, potentially introducing upward bias into aggregate statistics (Hayden and Christy, 2023). The black solid lines denote exact 1:1 agreement. See the SI, Section S4 for Stage 1 and Stage 2 satellite- and team-specific results.

## 2.3   Qualitatively assessing detection performance in the field

The smallest emission detected by each team gives a rough upper bound on the lower detection capabilities of each instrument, at least in a desert environment with a known release location. We compare these smallest detected emissions with previous estimates of lower detection capabilities of each satellite. The smallest emission detected was 0.0332 [0.0328, 0.0336] t/h, identified by Maxar using WorldView-3, shown in Figure 6. Kayrros also detected an emission below 0.1 t/h using WorldView-3. This is consistent with previous estimates of lower detection capabilities, with Sánchez-García et al. detecting an emission estimated at ~0.040 t/hr in Turkmenistan using WorldView-3 (Sánchez-García et al., 2022).

Orbio Earth, Maxar, and GHGSat all detected a 1.19 [1.15, 1.23] t/h emission using Sentinel-2, with errors ranging from -8% to +170%. Orbio Earth detected a 1.05 [0.99, 1.10] t/h emission to within ±47%. These emissions are 15-25% below the smallest emission detected using Sentinel-2 in any previous satellite controlled methane release test, and consistent with simulation-based

estimates (Sherwin et al., 2023b; Gorroño et al., 2023). The story is similar for LandSat 8/9, with
the smallest detected emission at 1.39 [1.34, 1.43] t/h. This is also slightly below estimated lower
detection capabilities in the literature (Jacob et al., 2022).
The smallest emission detected via PRISMA was 0.414 [0.410, 0.417] t/h smaller than the 0.5-
2.0 t/h estimated by Guanter et al. as PRISMA's lower detection threshold (Guanter et al., 2021).
The smallest detected emissions for the remaining satellites are 1.10 [1.06, 1.13] t/h for EnMAP,
1.26 [0.26, 2.26] t/h for GF5, and 1.03 [0.98, 1.09] t/h for ZY1. However, given that the
technical characteristics of these three satellites are similar to PRISMA, they can likely be used
to detect emissions below 1 t/h, at least under favorable environmental conditions (Jacob et al.,
2022; Roger et al., 2023).
GHGSat correctly detected and quantified the only nonzero release for which GHGSat-C
collected data and passed quality control, which was 0.401 [0.399, 0.404] t/h, roughly double the
smallest release GHGSat quantified using the same satellite system in (Sherwin et al., 2023b).
GHGSat's lower detection threshold is estimated at 0.1-0.2 t/h (Jacob et al., 2022). HJ2B was not
tasked during any active releases, meaning that future testing is needed to assess its detection
capabilities.
In practical applications for global remote sensing, teams have only limited information about
the location of possible sources and their likelihood of emitting at visible levels. As a result, it is
possible that the known-location experimental design applied here may have allowed teams to
artificially boost detection sensitivity to levels that would be difficult to achieve in general
practice.
To qualitatively assess this possibility, all teams were required to submit methane retrieval field
images for all submitted estimates, including both detections and non-detections. In all cases,
teams submitted full-scene retrieval fields in a 2x2 km box around the release location. For
detected emissions, teams also submitted masked plume images, overlaying the estimated
methane plume above an optical image of the background location. See the SI, Section S4 for all
such images.
We highlight selected images in Figure 6 to showcase issues related to spectral artifacts, e.g.
apparent methane enhancements due to water bodies, clouds, or roads, that we were not able to
quantitatively address in this study. The GHGSat images, shown at a contrast-enhancing
narrower color scale of 0-0.2 ppm instead of this study's standard 2 ppm, show that for the
November 8[th] retrieval of the 0.401 [0.399, 0.404] t/h release, there are pixel clusters with
enhancements of comparable magnitude outside of the release area. However, these
enhancements are concentrated along ground features such as a water body southwest of the site
and a highway north of the site, confirmed in Google Maps imagery and WorldView-3 optical
images in the SI, Section S4. As a result, automated or manual intercomparison of the spatial
overlap of apparent methane enhancements and ground features visible in optical imagery could
plausibly help differentiate between such signal artifacts and true emissions. In some cases, it
may be possible to use measurements in which there is no evidence of a methane emission, e.g.
the November 16[th] measurement (in which GHGSat correctly determined the absence of methane
in a single-blind manner), to gain additional information into ambiguous cases. Artifacts such as
the water feature may consistently appear across retrievals, which could suggest that they are not
true methane enhancements. Furthermore, GHGSat flagged the water body in both retrievals as a
potential artifact, indicating that it would likely have been possible to correctly identify only the
true methane emission in the November 8th scene even without a reference image with no
methane.

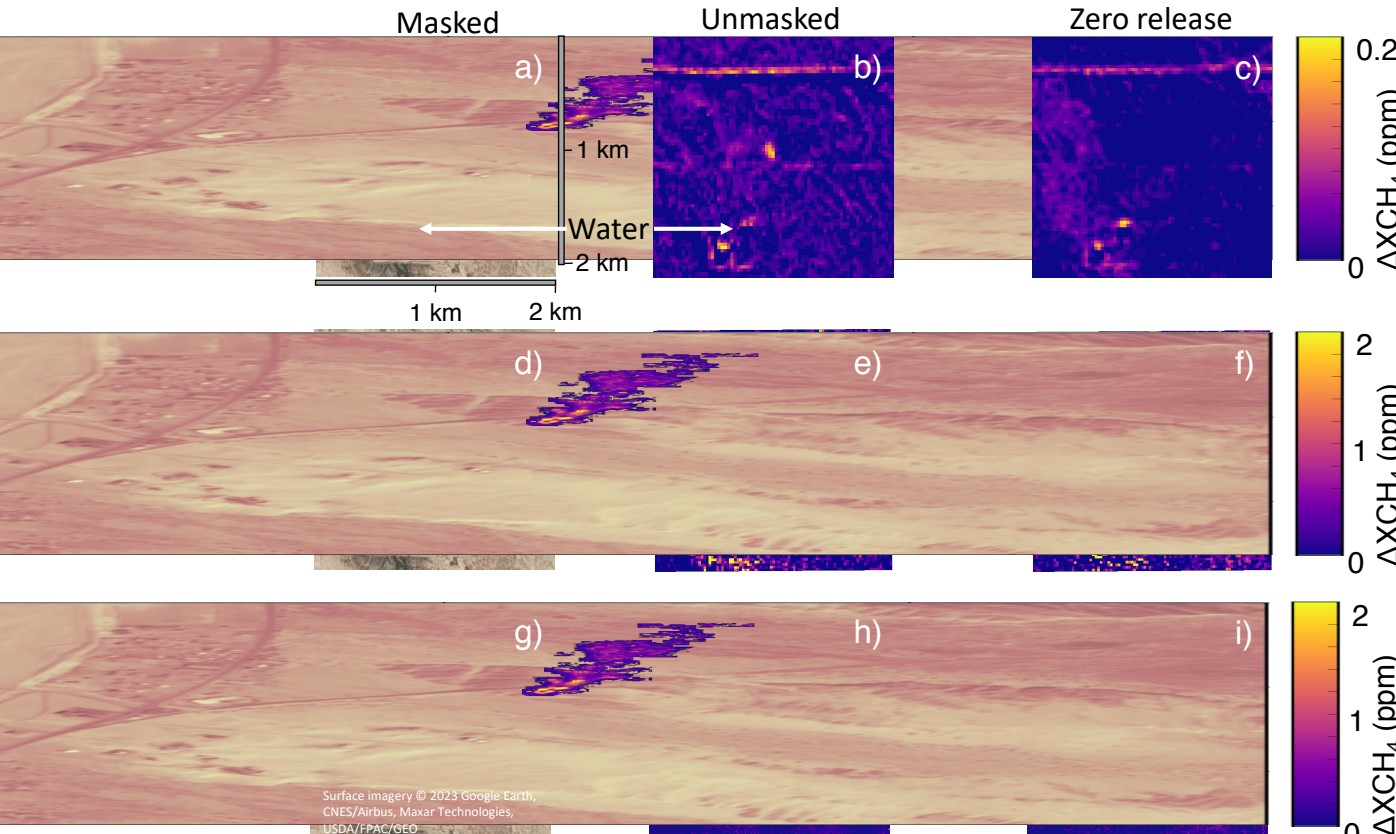

Figure 6. Masked and unmasked retrievals for selected emissions. In each case, the unmasked retrieval in the middle
column appears to contain artifacts of similar intensity and shape to the masked emission. However, the emission
may be more distinguishable from artifacts after intercomparison with ground features revealed through optical
imagery, e.g. the water body southwest of the release site, and intercomparison with a reference day with zero
emissions, as in the right column. Note that the GHGSat retrievals use a higher-contrast scale of 0-0.2 ppm. For See
the SI, Section S4 for GHGSat images using the standard 0-2 ppm scale applied for most retrieval images in this
study. Cloud-free surface imagery © 2023 Google Earth, CNES/Airbus, Maxar Technologies, USDA/FPAC/GEO.
Sentinel-2 imagery is significantly noisier than most other tested satellites. The November 18th
Kayrros retrieval in Figure 6 shows noticeable enhancements, comparable in intensity to the true
emission, along the water feature and the highway, as well as northwest of the release site. In
such a noise environment, knowledge of the emission location and access to images known not
to contain emissions, such as panel f) may assist in correct identification of the true emission.
See the SI, Section S4 for all masked and unmasked retrieval images from all satellites.
Maxar correctly detected emissions as small as 0.0332 [0.0328, 0.0336] t/h using their
WorldView-3 satellite on October 29th. Interestingly, their retrieval algorithm does not appear to
introduce high-concentration artifacts over the water body (although that is not the case for all
teams analyzing WorldView-3 data, as shown in the SI, Section S4). The full retrieval image for
the October 29th retrieval shows concentration enhancement artifacts of comparable magnitude to
the correctly-detected emission at several points in the image. However, these artifacts are
largely conformal with surface features visible in optical imagery.
In addition to the known location, Maxar tasked its WorldView-3 satellite without notice to the
Stanford team on November 24th, a holiday in the United States. These data were shared with all
teams, but Maxar did not submit an unmasked image for the November 24th retrieval, although
they did for the zero-emission November 5th retrieval, shown in panel i). As a result, Maxar and
all other participating teams were able to compare satellite data from active testing days with
data that they knew very likely did not contain methane enhancements. As a result, these teams
had information in addition to the known release location that would not necessarily be available
in the field. As a result, we cannot definitively conclude from this study whether Maxar or other
teams would successfully identify emissions as small as 0.0332 [0.0328, 0.0336] t/h in the field.
Future testing, likely with multiple potential source locations, is needed to more rigorously
assess field-realistic detection limits of all satellites tested in this study.

## 2.4    The role of clouds

Because water vapor is highly absorptive in the methane-active infrared frequencies targeted by
all nine methane-sensing satellites tested in this study, cloud cover can impede or prevent valid
satellite-based methane measurements. Although our Arizona test site was selected in part due to
its arid, relatively low-cloud climate, periodic cloud cover occurred to varying degrees
throughout the testing period.
The treatment of clouds varied across teams, with some filtering images due to cloud cover more
aggressively than others. LARS filtered the November 22nd WorldView-3 retrieval, shown in
Figure 7a, noting "the image is cloudy but we see some enhancement." Kayrros and Maxar
correctly detected the 0.433 [0.430, 0.436] t/h emission for the same measurement, while NJU
reported a non-detection.

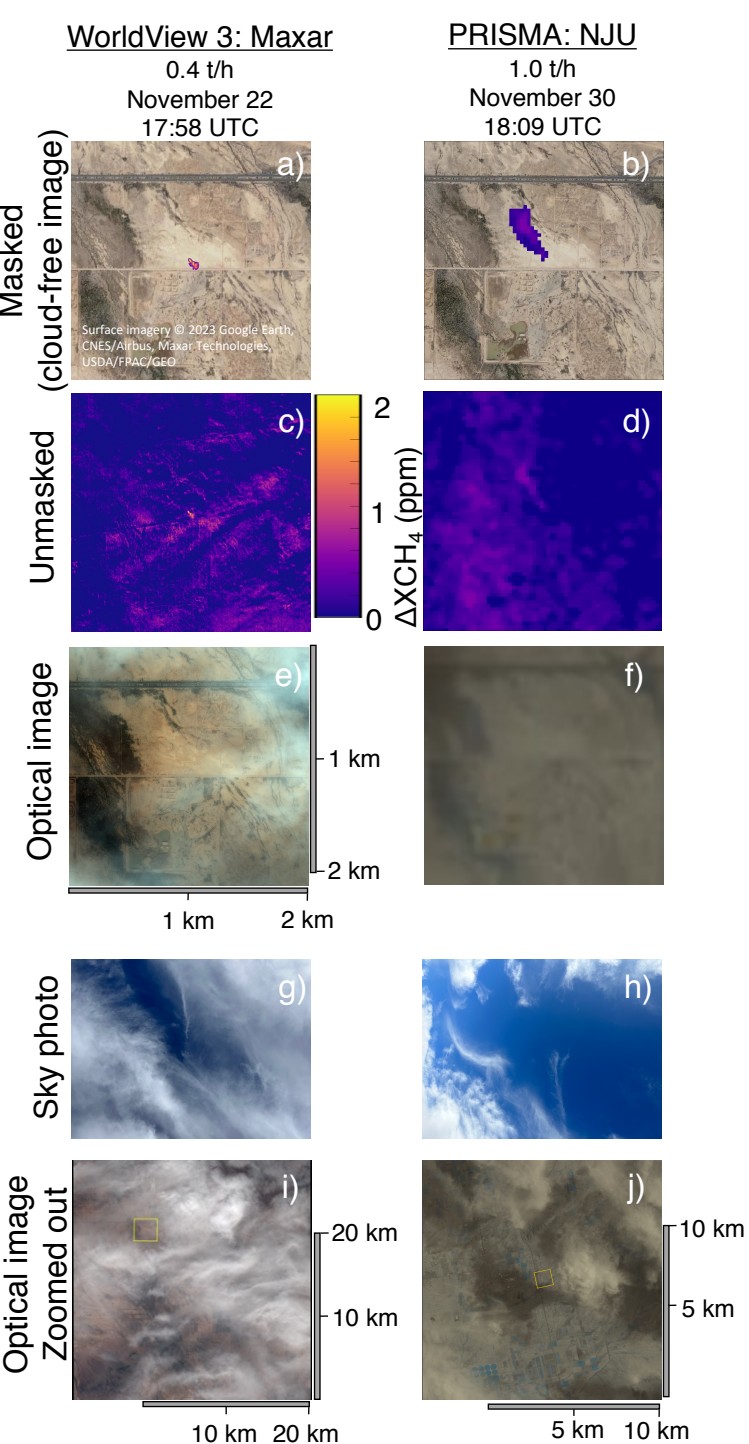

Figure 7. Cloudy days with successful methane detections. a) and b) show masked methane emissions from
WorldView-3 and PRISMA above a cloud-free standard background © 2023 Google Earth, CNES/Airbus, Maxar
Technologies, USDA/FPAC/GEO. c) and d) show corresponding unmasked images. e) and f) show optical images
of the same 2x2 km scene collected by each satellite. g) and h) show photographs of the sky, taken by Stanford
researchers on smartphones at the time of each overpass. i) and j) show zoomed-out versions of the optical images
shown in e) and f), with different length scales than the other panels.
This highlights that accurately interpreting the results of field measurements from each of these
teams requires an understanding of both detection performance and data filtering processes as a
function of cloud cover.
Stanford researchers took photographs of the sky coincident with most satellite overpasses to
document cloud cover, shown in full in the SI, Section S4. The photograph for the November
22nd WorldView-3 overpass, Figure 7g, appears to show significant thick cloud cover. However,
analysis of optical WorldView-3 imagery from this measurement, Figure 7e, shows that the area
immediately above the test site was relatively cloud-free even though the broader area was
experiencing significant cloud cover, shown in Figure 7i.
Analysis of the November 30th PRISMA measurement, shown in the second column of Figure 7
adds further nuance to the question of cloud cover. The sky photograph in Figure 7h shows the
presence of thin clouds. However, the optical image collected by PRISMA in Figure 7f shows no
clouds within the 2x2 km square surrounding the release site. The photographed clouds are only
visible in the larger, 14x14 km image in Figure 7j, which demonstrates that clouds are too far
away from the release site to interfere with the 0.98 [0.88, 1.08] t/h methane plume, which was
correctly detected by LARS, NJU, and Maxar.
These two cases demonstrate that only limited information regarding cloud cover can be
determined from single-frame sky photographs taken from the ground. This is particularly true
without clear orientation information, which is not available for the smartphone-based
photographs used in this study.
Figure 8 shows sky photographs of all dates with valid or operator-filtered GHGSat
measurements. Both days with valid measurements, one true positive and one true negative, were
essentially cloudless, as shown in Figure 8d-e. In addition, GHGSat filtered three retrievals due
to clouds. Of the three days filtered due to cloud cover, one was fully overcast (Figure 8c), while
two had thin clouds, shown in Figure 8a-b, also noted in the GHGSat report for those days. As
demonstrated above, it is difficult to determine from these sky photographs alone where these
clouds were in relation to the release site.

# GHGSat-coincident sky photos

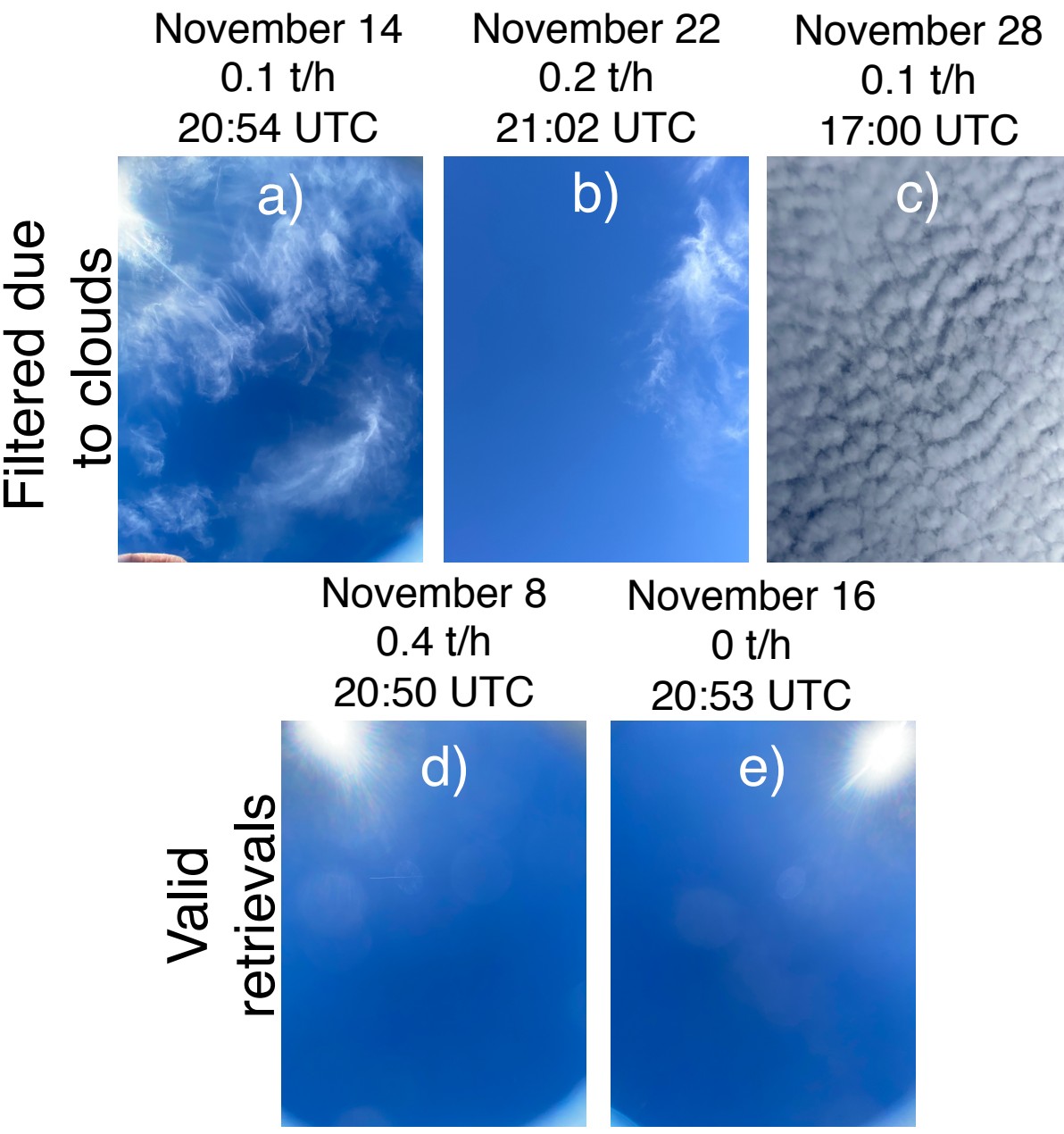

Figure 8. Ground-perspective sky photos for GHGSat-C measurements. a-c) correspond to measurements filtered due to cloud cover. d) and e) correspond to valid retrievals, including one true positive detection and one true negative non-detection. GHGSat-C satellites do not collect optical imagery, making it difficult to directly compare ground-perspective photographs with satellite-perspective optical imagery.

GHGSat did not submit unmasked retrieval images for operator-filtered measurements (these images were requested from all teams, but were not required as a condition of participation in this test). Furthermore, GHGSat does not collect optical imagery in visible frequencies, so none

could be submitted. As a result, we can draw only limited conclusions about the role of cloud
cover in GHGSat's ability to conduct valid measurements with the GHGSat-C satellite model.

Future satellite-focused controlled methane release tests should further investigate the role of
cloud cover. This should include conducting testing in cloudier locations. In addition, sky
photographs should be replaced by or supplemented with passively-collected time series of
panoramic, georeferenced sky time series, e.g. using a fisheye camera, e.g., as used in solar
forecasting systems (Sun et al., 2018). This, together with optical images collected by satellites
(when available), will allow a more systematic evaluation of the capabilities of the tested
systems as a function of cloud cover. Such analysis should include assessment of the effect of
clouds on detection sensitivity and quantification performance, as well as their role in preventing
collection of valid measurements. These cloud-informed performance findings will be
indispensable in regional analysis of satellite-based methane remote sensing data, including its
incorporation into emissions inventories.

## 3 Discussion

This work demonstrates that all tested satellites are capable of detecting and quantifying methane
emissions. All eight satellites given the opportunity detected methane emissions, with overall
quantification accuracy similar, in percent terms, to aircraft-based methane sensing systems. This
highlights the large suite of satellite-based tools available to detect and quantify methane point
sources across the globe.

Detection limits appear to improve with smaller swath width and pixel size, and with higher
spectral resolution. Global-coverage satellites such as LandSat 8/9 and Sentinel-2, with swaths of
185 and 290 km, respectively, and spectral resolution 20-650 times coarser than the
hyperspectral instruments (EnMAP, PRISMA, GF5, ZY1, HJ2B, and GHGSat), have higher
detection limits. See the SI, Section S2 for additional discussion of spectral resolution. Our
results are consistent with (Gorroño et al., 2023), whose simulation-based approach suggests that
such instruments have a best-case minimum detection limit of roughly 1 t/h. Targeted satellites
with swaths of 30-60 km, including EnMAP, GF5, PRISMA, and ZY1 (EnMAP, 2023; Liu et al.,
2019; OHBI, 2022; Song et al., 2022), all reliably saw emissions of ~1 t/h. Of these, only
PRISMA has had the opportunity to be tested with emission fluxes below 1 t/h, correctly
detecting 0.413 [0.410, 0.417] t/h, the smallest emission given to PRISMA. GHGSat correctly
detected 0.401 [0.399, 0.403] t/h, with quantification accuracy within ±20%, using their
GHGSat-C-series satellite, with a swath width of 12 km. Estimates for smaller emission sizes
were filtered due to clouds, but in previous testing GHGSat successfully detected an 0.197
[0.187, 0.208] t/h emission and quantified it with similar accuracy, suggesting that the system
may be capable of seeing emissions even smaller than 200 kg/h.

Maxar successfully detected emissions as low as 0.0332 [0.0328, 0.0336] t/h using the
WorldView-3 satellite, with swath width 13.1 km. Two teams successfully detected emissions
below 0.1 t/h using WorldView-3, while two teams applied more conservative criteria and
detected only emissions above 0.5 t/h. Although Maxar has a coarser spectral resolution than
hyperspectral instruments, its very high spatial resolution enables heightened sensitivity.

In the high-emission New Mexico Permian basin oil and natural gas system, using 2019 emission
levels, a comprehensive measurement campaign with a constellation of satellites detecting all
emissions above 1 t/h would find 20% of emissions from oil and gas well sites, rising to 62% for
a satellite detecting emissions above 0.2 t/h, and 83% above 0.03 t/h (Sherwin et al., 2023b).
These fractions are upper-bound estimates both because near-real-time comprehensive coverage
would be challenging for satellite systems and because the underlying emission size distribution
estimate may be conservative for emissions below roughly 50 kg/hr (Sherwin et al., 2023b). In
lower-emitting basins such as the Denver-Julesburg, each of these systems would see a much
smaller fraction of total emissions, highlighting the need for a variety of technology approaches,
tailored to regional system characteristics (Sherwin et al., 2023b).
Note that the detection results presented in this paper reflect system performance with a known
source location under favorable desert climate conditions. These results may not translate to field
performance in different environments and with less foreknowledge about the location of
possible sources.
Unmasked methane retrieval fields, submitted by all teams, suggest that achievable detection
limits may be higher in practice for some satellites. In some cases, these images contain
background artifacts with estimated methane enhancements comparable in magnitude and
qualitatively similar in shape to the detected methane plumes. However, in many of these
retrieval fields, particularly for larger emissions, the true methane plume is unambiguous. It is
noteworthy that some teams correctly flagged likely background artifacts in blinded submissions,
but such georeferenced quality flagging was not required of all participating teams, although
doing so may be advisable in future tests.
The role of surface features, such as water bodies, in creating apparent methane enhancements
should be explored further. For example, the retrieval field for the 0.401 [0.399, 0.403] t/h
GHGSat measurement shows an apparent methane enhancement over a water body that is similar
in magnitude to the detected plume. However, if this is a known characteristic of the algorithm,
then such artifacts could be automatically or manually filtered out, leaving only the clear
methane plume at the release site. The water body appears as a flagged region in all data reported
by GHGSat, indicating that their system is capable of identifying potential confounding factors
such as water bodies and differentiating any resulting artifacts from true methane emissions.
Clouds add several levels of complexity to satellite-based methane sensing. The water vapor in
clouds interferes with the frequencies all tested satellites use to identify methane enhancements.
Heavy cloud cover essentially prevents valid satellite-based methane sensing. This test
demonstrates that it is possible in some circumstances to detect and quantify methane emissions
even in the presence of nearby patchy or thin clouds. However, it is unclear in some cases
whether these detected emissions would have been distinguishable from background noise, e.g.
artifacts caused by clouds or highly reflective/absorptive surface features, in the absence of a
known source location and reasonable anticipation of the presence of an emission due to an
ongoing test.
Different teams employed different filtering criteria. GHGSat excluded all GHGSat-C
measurements with cloud cover. Maxar and Kayrros used WorldView-3 to successfully detected
a 0.433 [0.430, 0.436] t/h emission on a cloudy day on November 22$^{nd}$, while LARS filtered the
measurement due to clouds and NJU reported a non-detection.
Future testing should characterize the cloud conditions under which valid point-source methane
measurements can and cannot be conducted with each satellite-based system. In addition, future
work should characterize the effect of partial cloud cover on detection and quantification
performance. Understanding these two factors will be critical when interpreting the results of
large-scale satellite-based methane measurement campaigns, which will inevitably encounter
interference from clouds. Cloud cover varies widely across oil and gas-producing regions, with
limited clouds in arid areas such as the Permian basin in Texas and New Mexico, and significant
cloud cover in more temperate producing regions such as the Appalachian basin in the eastern
United States and the Williston basin in the midwestern United States (NASA, 2023).
It is noteworthy that even under cloud-free conditions, a targeted satellite overpass is not
guaranteed to produce valid data. Errors in tasking software, as well as onboard hardware upsets
can prevent valid data collection. The incidence of both in this paper may not be representative
of field performance for the tested technologies. Additional data collection, ideally from field
data, would be needed to accurately quantify the incidence of data collection failure, and further
location-specific analysis of cloud trends would be needed to understand the impact of cloud
cover on satellite data collection capabilities in a specific area.
Wind speed remains a major driver of uncertainty in satellite-based methane point source
quantification. Moving from wind reanalysis data to in situ wind measurements substantially
reduces scatter around the line of best fit, as was also the case in other work from the same group
(Sherwin et al., 2023b). In addition, in situ wind measurements show considerable temporal
variability in wind speed and direction over the multi-minute timescales most relevant to plume
formation.
In the field, winds are generally only available from reanalysis data, which capture temporal,
spatial, and directional variability with much lower fidelity than on-the-ground wind
measurements. Advances in the spatial and temporal fidelity of wind reanalysis products, as well
as their accuracy, could help improve methane remote sensing. In addition, it may be possible to
entirely eliminate reliance on wind speed, e.g. by inferring emission rate information solely from
plume shapes as in reference (Jongaramrungruang et al., 2022).
It is important to note that conducting this test did require the release of considerable amounts of
methane into the atmosphere. We estimate total emissions from the satellite testing discussed in
this paper at 7.7 t($CH_4$), discussed further in the SI, Section S1.5. However, this pales in
comparison with anthropogenic emissions occurring across the globe. Lauvaux et al. identify
over 1000 emission sources across the world emitting at least 7.7 t($CH_4$) every hour, in some
cases over 50 times as much every hour (Lauvaux et al., 2022). If this work assists in
accelerating mitigation of even one of these emissions by even a single hour, e.g. by ensuring
key decision-makers view satellite-based methane detection and quantification as reliable, we
will have broken even from a methane emissions perspective.

The findings presented here demonstrate that at least eight distinct satellite systems from three
continents are capable of detecting methane point sources of 1.5 t/h or less. Furthermore, this
study more systematically probes lower detection limits of these systems, two teams detecting
emissions below 0.1 t/h, the first time to our knowledge that such performance has been
demonstrated in a single-blind test of satellite-based methane sensing systems.
These satellites can play an important role in reducing methane emissions through existing
regulatory pathways, both in the United States and internationally. The US Environmental
Protection Agency's proposed update to rules governing methane emissions from oil and natural
gas production includes a super-emitter response program, in which approved third-party data
providers can flag identified emissions above 0.1 t/h, obliging operators to investigate further
and, if necessary, take action to halt any further emissions (EPA, 2022). A proposed update to
the EPA Greenhouse Gas Reporting Program also includes a new category of "Other large
release" for inclusion in company emissions reports (EPA, 2023). The Methane Alert and
Response Systems, part of the United Nations' International Methane Emissions Observatory,
uses vetted satellite data to notify governments, and in some cases operators, of large emissions
detected by satellite, with the aim of mitigating these emissions (IMEO, 2023). The eight
satellite systems tested with at least one nonzero emission in this study can provide high-quality
data to each of these programs.
In coming years, the Carbon Mapper and MethaneSAT systems will launch, alongside additional
satellites in some of the constellations tested here (Jacob et al., 2022). The airplane-mounted
precursors to both the Carbon Mapper and MethaneSAT systems have conducted substantial
single-blind testing of their point-source detection and quantification capabilities (Rutherford et
al., 2023; Chulakadabba et al., 2023; El Abbadi et al., 2023), but the satellites will require
additional tests. Furthermore, the NASA Earth Surface Mineral Dust Source Investigation
(EMIT) system, which launched shortly before our testing began (Wang and Lee, 2022), has
already reported detecting methane emissions in the field and should be tested, along with the
HJ2 system, in future single-blind controlled methane releases.
The tools exist for multi-lateral global methane monitoring efforts, with satellites from multiple
countries and continents able to independently assess emissions from regions of interest. The
single-blind test conducted here is a step toward ensuring that stakeholders across the world have
confidence in the methane emissions these satellite systems find at oil and gas facilities, landfills,
coal mines, and other emitting infrastructure. This will help satellites achieve their potential to
not only detect and quantify large methane emissions, but to inspire meaningful action to reduce
emissions of this powerful greenhouse gas.

**4    Data and code availability**
All data and code required to reproduce the figures and analysis in this paper are available at:
https://github.com/sahar-elabbadi/SU-Controlled-Releases-2022. Underlying spectral imagery
will not be made directly available through this study, but for many satellites tested in this study
these spectral data can be acquired either for free or for purchase for via platforms discussed in
the SI, Section S2.

## 5  Abbreviations

| | |
|---|---|
| ADED | Advancing Development of Emissions Detection |
| ASI | Italian Space Agency |
| CNG | Compressed Natural Gas |
| EMIT | Earth Surface Mineral Dust Source Investigation |
| EnMAP | Environmental Mapping and Analysis Program |
| GF5 | Gaofen 5 |
| GSC | GHGSat-C (satellite) |
| HJ2 | Huanjing 2 |
| IME | Integrated Mass Enhancement |
| kg/h | Kilograms per hour |
| LARS | Land and Atmosphere Remote Sensing |
| LS | LandSat |
| METEC | Methane Emissions Technology Evaluation Center |
| NASA | National Aeronautics and Space Administration |
| NJU | Nanjing University |
| NOAA | National Oceanographic and Atmospheric Administration |
| OHB | Orbitale Hochtechnologie Bremen |
| PRISMA | PRecursore IperSpettrale della Missione Applicativa |
| UPV | Universitat Politècnica de València |
| USGS | United States Geological Survey |
| SRON | Stichting Ruimte Onderzoek Nederland |
| SWIR | Short-wave Infrared |
| TROPOMI | TROPOspheric Monitoring Instrument |
| t/h | Metric tons per hour |
| VNIR | Visible to Near Infrared |
| WAV-P | Wide-Angle Fabry-Perot |
| WV3 | WorldView-3 |
| ZY1 | Ziyuan 1 |

## 6  Acknowledgments

We acknowledge C. de Franchis, C. Giron, and A. Groshenry of Kayrros; and Z. Mouton, W. Kingwill, and R. Huppertz from Orbio Earth for their participation in this test. A. Esparza, L. Clark-Squire, J.F. Gauthier, M. Girard, D. Jervis, R. Mattson, J. McKeever, A. Newhook, and M. Turenne of GHGSat; J. Gorroño Viñegla, J. Roger Juan, and L. Guanter Palomar of LARS; Chen H., Li F., and Zhang H. of NJU; A. Hayden, J. Jonik, and J. Christy of Maxar both for participating in the test and for coordinating tasking and data sharing from key satellites. We acknowledge the German Aerospace Center and the Italian Space Agency for tasking the EnMAP and PRISMA satellites, respectively. Rawhide Leasing (Dana Walker) and Volta Fabrication (Mike Brandon, Walt Godsil, S.M., Merritt Norton) provided indispensable operational, logistical, and planning support for the experiment.

## 7 Author contributions

Conceptualization – EDS and ARB. Methods – EDS, SHE, YC, JSR, and ARB. Software – EDS, PMB, ZC, and SHE. Validation – EDS. Formal analysis – EDS, PMB, ZZ, and SHE. Investigation – EDS and SHE. Resources – ARB. Data curation – EDS, PMB, YC, ZZ, ZC, JSR, and SHE. Writing: original draft – EDS. Writing: review and editing – All authors. Supervision – EDS and ARB. Project administration – EDS, SEA, and ARB. Funding acquisition – EDS, SHE, ARB. This work was supported by the US Department of Energy, Office of Science through contract DE-AC02-05CH11231 between Lawrence Berkeley National Laboratory and the US Department of Energy. The US government retains, and the publisher, by accepting the article for publication, acknowledges that the US government retains, a non-exclusive, paid-up, irrevocable, world-wide license to publish or reproduce the published form of this manuscript, or allow others to do so, for US government purposes.

## 8 Supplementary information available

The online version contains supplementary material.

## 9 Funding sources

This study was funded by: The Environmental Defense Fund, the Global Methane Hub, the International Methane Emissions Observatory, and the Stanford Natural Gas Initiative, an industry consortium that supports independent research at Stanford University.

## 10 Competing interests

ARB serves on the Science and Measurement advisory committee of Carbon Mapper, which plans to launch a methane-sensing satellite. YC and ZZ previously worked as interns at Carbon Mapper. The remaining authors have no competing interests to declare.

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
