# Peer review of "Title: Single-blind test of nine methane-sensing satellite systems from three continents"

_EGUsphere, 2023_

## Author Response (AR1)

Dear Editor,

Thank you for your consideration at Atmospheric Measurement Techniques. We include itemized responses to the very helpful editorial and reviewer comments, which provided a further opportunity to refine and streamline the paper. We hope these revisions will adequately address the concerns raised in review.

We reproduce reviewer comments below and include our responses in indented text, with updates to the manuscript reproduced in doubly-indented text in blue. We have also included a version of the manuscript with tracked changes.

As requested by the editor's office, we have updated the citation formatting and supplementary information table and figure numbering to match AMT guidelines.

Thank you once again for this opportunity. We look forward to your and the reviewers' response.

Kind regards
Evan
* * *
AMT Satellite testing paper reviewer responses
Evan Sherwin

You are kindly asked to individually respond to all referee comments (RCs) that have not yet been answered (marked in red). You can choose between posting a new author comment (AC) and co-listing an existing one in response to an RC. You are also invited to respond to other discussion contributions, if applicable.

**Status**: final response (author comments only)
**RC1**: 'Comment on egusphere-2023-1541', Anonymous Referee #1, 19 Aug 2023  reply
Sherwin et al. evaluated the performance of satellite techniques for detecting and quantifying methane emissions through a single-blinded test. The test is well designed and carried out, providing timely and objective information critical for stakeholders and potential users. The technical complications (e.g., known vs. unknown location, clouds) are also well discussed. I appreciate that the authors documented the study from coordination to implementation in a great detail. I'd recommend the publication of this manuscript after the following comments are addressed.
Main comments
An important conclusion is that "quantification performance of satellite techniques approaches aircraft accuracy". But this statement is not elaborated. Only a brief comparison with values from previous studies is made. In these previous tests, the tested flux may have very a different distribution than that in this study. I wonder how a different distribution of tested fluxes may affect the conclusion. This key finding may
be better established if the analysis can be done more carefully. For example, the
concern mentioned above may be addressed with an evaluation of the quantification
accuracy for subsets with a similar distribution of metered flux. Moreover, this
comparison in quantification performance is not the full picture and may mislead
readers. Detection performance (detection limit) of satellite and aircraft technologies
should also be compared, in addition to quantification performance.

**Authors:** We have updated this section, now entitled "**Reliable overall**
**quantification performance**" to more clearly convey key points and avoid
potential confusion.
We now highlight in the first and second paragraphs of this section:

**Manuscript, L305:** "However, the best-fit line across all satellite measurements, any one of
which may have substantial quantification error, is largely unbiased, with a slope close to the
ideal value of 1 (which would denote perfect agreement on average).

[Figure]

Figure 1. Methane quantification performance by satellite and team. Metered
emissions compared with single-blind estimates for each overpass with successfully
reported data, with 95% X and Y confidence intervals. a) Fully blind stage 1 results
using modeled wind speed estimates. Note one Sentinel-2 estimate exceeds the y-
axis limit at 6.6 t($CH_4$)/h. b) Stage 2 results using on-site 10 m wind speed and
direction measurements. LARS WorldView-3 quantification estimates are excluded
from the main analysis, as stage 1 estimates were submitted after wind data had
been unblinded to a member of the LARS team not involved in analyzing
WorldView-3 data, while corresponding stage 2 estimates were submitted after release volumes were unblinded. Note that Maxar submitted PRISMA estimates for
stage 2 only. The grey dashed lines represent an ordinary least squares fit with the
intercept fixed at zero, with slope and uncentered $R^2$ displayed. Maxar has since
determined that its estimates were likely artificially high, potentially introducing
upward bias into aggregate statistics (Hayden and Christy, 2023). See the SI,
Section S4.2 for a version of this plot excluding Maxar, which shows overall
improvement in both slope and $R^2$. The black solid lines denote exact 1:1
agreement. See the SI, Section S4 for satellite- and team-specific results."

In Page 15, the smallest detected emissions for each satellite are reported as a metric
of detection performance. This information may be misleading. For example, both ZY1
and GF5 are only tested once. They are not tested with smaller fluxes, which is different
from other missions that are tested with a range of fluxes. So, the "smallest detected
emissions" from these missions should be interpreted differently. In addition, I wonder
if it is possible to perform a more rigorous analysis of "observed detection limit". This
should be possible for missions that are tested with a range of fluxes. And a
comparison with theoretical detection limits (e.g., as reported in Jacob et al. 2022 ACP
review) should bring additional insight.

**Authors:** We have expanded our discussion of the detection capabilities of
these instruments, combining our empirical findings for the controlled releases
we conducted with other existing theoretical and simulation literature to present
the most comprehensive picture we can of the likely detection capabilities of
each system tested in this study.

**Manuscript, L469:** "The smallest emission detected by each team gives a rough
upper bound on the lower detection capabilities of each instrument, at least in a
desert environment with a known release location. We compare these smallest
detected emissions with previous estimates of lower detection capabilities of each
satellite. The smallest emission detected was 0.0332 [0.0328, 0.0336] t/h,
identified by Maxar using WorldView-3, shown in **Error! Reference source not**
**found.**. Kayrros also detected an emission below 0.1 t/h using WorldView-3. This
is consistent with previous estimates of lower detection capabilities, with
Sánchez-García et al. detecting an emission estimated at ~0.040 t/hr in
Turkmenistan using WorldView-3 (Sánchez-García et al., 2022).

Orbio Earth, Maxar, and GHGSat all detected a 1.19 [1.15, 1.23] t/h emission
using Sentinel-2, with errors ranging from -8% to +170%. Orbio Earth detected a
1.05 [0.99, 1.10] t/h emission to within ±47%. These emissions are 15-25% below
the smallest emission detected using Sentinel-2 in any previous satellite controlled
methane release test, and consistent with simulation-based estimates (Sherwin et
al., 2023; Gorroño et al., 2023). The story is similar for LandSat 8/9, with the
smallest detected emission at 1.39 [1.34, 1.43] t/h. This is also slightly below
estimated lower detection capabilities in the literature (Jacob et al., 2022).

The smallest emission detected via PRISMA was 0.414 [0.410, 0.417] t/h smaller
than the 0.5-2.0 t/h estimated by Guanter et al. as PRISMA's lower detection
threshold (Guanter et al., 2021). The smallest detected emissions for the
remaining satellites are 1.10 [1.06, 1.13] t/h for EnMAP, 1.26 [0.26, 2.26] t/h for
GF5, and 1.03 [0.98, 1.09] t/h for ZY1. However, given that the technical
characteristics of these three satellites are similar to PRISMA, they can likely be
used to detect emissions below 1 t/h, at least under favorable environmental
conditions (Jacob et al., 2022; Roger et al., 2023).

GHGSat correctly detected and quantified the only nonzero release for which
GHGSat-C collected data and passed quality control, which was 0.401 [0.399,
0.404] t/h, roughly double the smallest release GHGSat quantified using the same
satellite system in (Sherwin et al., 2023). GHGSat's lower detection threshold is
estimated at 0.1-0.2 t/h (Jacob et al., 2022). HJ2B was not tasked during any
active releases, meaning that future testing is needed to assess its detection
capabilities."

Minor comments

Abstract: "...in which teams incorrectly claimed methane was present": The way this
sentence is written is kinda confusing that whether this clause defines false positives or
both false positives and false negatives.
**Authors:** This sentence now reads as follows

**Manuscript, L25:** "There were 41 false negatives, in which teams missed a true
emission, and 0 false positives, in which teams incorrectly claimed methane was
present."

The authors report values as mean [min, max]. But it is never explicitly defined whether
[min, max] presents +-SD, interquartile range, or 95% confidence interval. This should
be specified at the first appearance.

**Authors:** We evidently forgot to clarify at first use that uncertainty ranges
presented in this paper are 95% confidence intervals. We now clarify this at first
use.

**Manuscript, L70:** "this approach can be used to detect emissions ranging from
0.20 [95% confidence interval = 0.19, 0.21] metric tons of methane per hour
(henceforth t/h)"

Figure 3. What does the * sign besides the Gaofen-5 flux mean?
**Authors:** We now clarify the meaning of this asterisk in Figure 3 as follows,

**Manuscript, L259:** "*The Gaofen 5 measurement was rescheduled without notice to a time that happened to be one minute after releases had concluded for a different satellite, resulting in artificially high variability in the metered ground-truth flow rate."

Figure 4. It is mentioned in page 13 that Maxar data are excluded from the main result. To be consistent with this, the authors may want to add regression lines and statistics for data with Maxar excluded in Figure 4.

**Authors:** For consistency with prior studies, we continue to report only the linear fit to the single-blind estimates provided by all teams. However, we have added a version of Figure 4 that excludes the Maxar data to the SI, reproduced below. We now reference this figure in the caption of Figure 4, as well as in the manuscript. Unsurprisingly, excluding the Maxar results improves both the slope and R2 of the fitted line.

**Manuscript, Figure 4 caption:** "Maxar has since determined that its estimates were likely artificially high, potentially introducing upward bias into aggregate statistics (Hayden and Christy, 2023). See the SI, Section S4.2 for a version of this plot excluding Maxar, which shows overall improvement in both slope and $R^2$."

**Manuscript, L404:** "Excluding Maxar results (as in the SI, Section S4.2), the Stage 1 slope for all remaining teams falls to 0.897 [0.716, 1.078], with a Stage 2 slope of 1.010 [0.841, 1.180], almost perfect average agreement with metered values. These slopes are 21% and 19% below the respective estimates in which Maxar values were included."

**SI, Section S4.2:** "**S4.2. Regression results excluding Maxar**

[Figure]

Figure S1. Methane quantification performance by satellite and team, excluding estimates from Maxar, who realized after submission that their estimates were artificially high due to use of a deprecated spectral library. Metered emissions compared with single-blind estimates for each overpass with successfully reported data, with 95% X and Y confidence intervals. a) Fully blind stage 1 results using modeled wind speed estimates. Note one Sentinel-2 estimate exceeds the y-axis limit at 6.6 t(CH₄)/h. b) Stage 2 results using on-site 10 m wind speed and direction measurements. LARS WorldView-3 quantification estimates are excluded from the main analysis, as stage 1 estimates were submitted after wind data had been unblinded to a member of the LARS team not involved in analyzing WorldView-3 data, while corresponding stage 2 estimates were submitted after release volumes were unblinded. The grey dashed lines represent an ordinary least squares fit with the intercept fixed at zero, with slope and uncentered $R^2$ displayed. The black solid lines denote exact 1:1 agreement. See the SI, Section S4 for satellite- and team-specific results."

Page 12: The paragraph starting with "Wind can vary substantially ...". When wind information is revealed to the teams, are they informed of the distribution of the wind, or only the mean wind for the overpass? Is it possible that the distribution contain additional information that can help the teams further improve their estimates.

**Authors:** We now clarify that in the second stage of analysis, teams were provided with all in situ wind data at 1-second resolution,

**Manuscript, L164:** "In stage 2, Stanford provided 10 m wind speed and direction data from our on-site ultrasonic anemometer (shown in **Error! Reference source not found.**) at one-second resolution and teams were allowed to re-estimate emissions based on measured ground wind conditions rather than re-analysis products as in stage 1."

Page 14: "However, Orbio Earth successfully detected all Sentinel-2 releases above 0.010 t/h...". The statement is misleading. All Sentinel-2 detections are above 1 t/h.

**Authors:** We have updated this language to be clearer. The initial reason we chose this language is that there was one release at below 0.010 t/h during a Sentinel-2 overpass, and it would not be reasonable to count that as a false negative. This passage now reads as follows,

> **Manuscript, L446:** "However, Orbio Earth successfully detected all Sentinel-2 releases, except a release below 0.010 t/h (testing another technology), far below all estimates of the Sentinel-2 detection limit (Gorroño et al., 2023; Sherwin et al., 2023). These results highlight algorithmic variation across teams analyzing the same spectral data."

**RC2**: 'Comment on egusphere-2023-1541', Anonymous Referee #2, 31 Aug 2023  reply
The paper by Sherwin et al. evaluates satellites' performance in detecting and quantifying methane emissions from a fixed location. This work makes novel contribution to the literature as more satellites are being tasked for monitoring methane emissions. Kudos to the team - this is a very complicated field collaboration. I recommend accepting this paper with minor revisions as listed below:
1. The paper has referred to and cited many oil and gas methane studies. Assuming that one of the major usages of satellites is to monitoring methane emissions from oil and gas activities, could the authors add more context around how satellites can be deployed in the ever changing regulatory space? For example, whether these satellite could be used to monitor 'large releases' as defined by the new GHGRP rule.

**Authors:** We have added the following language to the introduction and discussion to highlight ways satellites can be incorporated into regulatory applications.

> **Manuscript, L787:** "These satellites can play an important role in reducing methane emissions through existing regulatory pathways, both in the United States and internationally. The US Environmental Protection Agency's proposed update to rules governing methane emissions from oil and natural gas production includes a super-emitter response program, in which approved third-party data providers can flag identified emissions above 0.1 t/h, obliging operators to investigate further and, if necessary, take action to halt any further emissions (EPA, 2022). A proposed update to the EPA Greenhouse Gas Reporting Program also includes a new category of "Other large release" for inclusion in company emissions reports (EPA, 2023). The Methane Alert and Response Systems, part of the United Nations' International Methane Emissions Observatory, uses vetted satellite data to notify governments, and in some cases operators, of large emissions detected by satellite, with the aim of mitigating these emissions (IMEO, 2023). The eight satellite systems tested with at least one nonzero emission in this study can provide high-quality data to each of these programs."

2. I'm a bit surprised by the comparison between aerial technology and satellite
performance (page 12 second to last paragraph). Were the emission rate tested for
aerial technologies much lower than that of satellite?

**Authors:** We have updated this section, now entitled "**Reliable overall**
**quantification performance**" to more clearly convey key points and avoid
potential confusion.
We now highlight in the first and second paragraphs of this section:

**Manuscript, L305:** "However, the best-fit line across all satellite measurements, any
one of which may have substantial quantification error, is largely unbiased, with a slope
close to the ideal value of 1 (which would denote perfect agreement on average).

[Figure]

Figure 2. Methane quantification performance by satellite and team. Metered
emissions compared with single-blind estimates for each overpass with successfully
reported data, with 95% X and Y confidence intervals. a) Fully blind stage 1 results
using modeled wind speed estimates. Note one Sentinel-2 estimate exceeds the y-
axis limit at 6.6 t($CH_4$)/h. b) Stage 2 results using on-site 10 m wind speed and
direction measurements. LARS WorldView-3 quantification estimates are excluded
from the main analysis, as stage 1 estimates were submitted after wind data had
been unblinded to a member of the LARS team not involved in analyzing
WorldView-3 data, while corresponding stage 2 estimates were submitted after
release volumes were unblinded. Note that Maxar submitted PRISMA estimates for
stage 2 only. The grey dashed lines represent an ordinary least squares fit with the
intercept fixed at zero, with slope and uncentered $R^2$ displayed. Maxar has since
determined that its estimates were likely artificially high, potentially introducing upward bias into aggregate statistics (Hayden and Christy, 2023). See the SI,
Section S4.2 for a version of this plot excluding Maxar, which shows overall
improvement in both slope and $R^2$. The black solid lines denote exact 1:1
agreement. See the SI, Section S4 for satellite- and team-specific results."
**Authors:** We have also added the following discussion of lower detection
capabilities:

**Manuscript, L371:** "In percent quantification error terms, this overall
performance approaches that of the satellites and teams tested in Sherwin et al.
2023, in which 75% of estimates fell within ±50% of the metered value,
demonstrating a relative error profile similar to that observed in aircraft-based
methane remote sensing technologies (albeit with minimum detection limits one
to three orders of magnitude larger) (Sherwin et al., 2023; El Abbadi et al., 2023;
Bell et al., 2022)."

3. Should success rate in generating usable datapoint be considered another metrics in
evaluate satellite performance? For example, when an aerial technology is deployed,
we expect to received usable data from their flyover. However, it seems like that's not
the case for satellite which could be results from uncontrollable factors such as cloud
coverage. Well not specific to any satellite, having a sense of the time period needed
for a satellite to produce usable data would be helpful in their deployment for constant
monitoring.

**Authors:** We now discuss the implications of the observed data collection
success rate in our study in the following paragraph.

**Manuscript, L748:** "It is noteworthy that even under cloud-free conditions, a
targeted satellite overpass is not guaranteed to produce valid data. Errors in
tasking software, as well as onboard hardware upsets can prevent valid data
collection. The incidence of both in this paper may not be representative of field
performance for the tested technologies. Additional data collection, ideally from
field data, would be needed to accurately quantify the incidence of data collection
failure, and further location-specific analysis of cloud trends would be needed to
understand the impact of cloud cover on satellite data collection capabilities in a
specific area."

**Authors:** In addition, we retain the following paragraph from the submitted
manuscript highlighting the need for additional testing to understand the impact
of cloud cover and different environments on data collection success rate.

**Manuscript, L738:** "Future testing should characterize the cloud conditions
under which valid point-source methane measurements can and cannot be
conducted with each satellite-based system. In addition, future work should
characterize the effect of partial cloud cover on detection and quantification performance. Understanding these two factors will be critical when interpreting the results of large-scale satellite-based methane measurement campaigns, which will inevitably encounter interference from clouds. Cloud cover varies widely across oil and gas-producing regions, with limited clouds in arid areas such as the Permian basin in Texas and New Mexico, and significant cloud cover in more temperate producing regions such as the Appalachian basin in the eastern United States and the Williston basin in the midwestern United States (NASA, 2023)."

4. If these satellite are being tested at active oil and gas facilities. How would the testing setup be different?

      **Authors:** We now highlight the main differences between our test facility and some of the most common types of oil and gas facilities at which satellites might be deployed.

            **Manuscript, L41:** "This experiment was designed to provide near-optimal conditions for methane-sensing satellites. In addition to the desert background, the site contained only equipment necessary to conduct controlled methane releases and test a suite of methane sensing technologies. The result is a significantly less complex scene than many oil and gas facilities, which will often contain multiple pieces of infrastructure such as wellheads, tanks, flares, and separators at production sites, and entire buildings with sophisticated machinery and piping at compressor stations and gas processing plants. More complex scenery can make methane remote sensing more challenging. Future work with scenes that more closely mimic industrial sites will help determine the associated differences in technology efficacy, if any."

**RC3**: 'Comment on egusphere-2023-1541', Anonymous Referee #3, 17 Sep 2023  reply
The manuscript by Sherwin et al. evaluates and documents in detail the capability to detect and measure methane point source emissions from point source satellite imagers that are currently in operation and have sufficient sensitivity to methane to detect emissions below 1.5 t/h. The information gathered in the document is highly important to clearly and transparently demonstrate the capability and limitations of these satellites and to guide stakeholders in assessing the reliability of these measurements. The manuscript is well written, and the experimental procedure is well detailed. I congratulate the authors and collaborators for the excellent work done here, and I would recommend the publication of this manuscript once the points and comments below are considered and corrected:
Major comments:
Either in Table 1 or in Section S2, the spatial resolution (pixel size) of each satellite should be indicated, an essential parameter to understand the detection and attribution capability of emissions from space. Furthermore, this parameter is mentioned at the beginning of the discussion, but readers do not have this information
in the manuscript.

**Authors:** We have added pixel size to Table 1, alongside the following context in
the manuscript,

**Manuscript, L127:** "These satellites range from high-sensitivity/narrow swath
to low-sensitivity/large swath, as illustrated in Table 1. Revisit time is also
anticorrelated with instrument sensitivity. The Sentinel-2 and LandSat 8/9
systems have estimated detection limits of roughly 1-5 t/h (Gorroño et al., 2023),
but each satellite in these constellations covers the bulk of the world's landmass
every 10-16 days with a swath of 185-290 km (USGS, 2022; ESA, 2021). GHGSat,
EnMAP, GF5, PRISMA, WorldView-3, and ZY1 are targeted "point-and-shoot"
systems, with higher resolution but narrower swaths of 12-60 km (ESA, 2022a, b;
Jervis et al., 2021; OHBI, 2022; EnMAP, 2023; Liu et al., 2019; Song et al.,
2022). Existing publicly available information does not specify whether HJ2 is
targeted or has global coverage, but its swath of 800 km suggests it is capable of
global coverage (Zhong et al., 2021). Pixel size also varies widely across
satellites, with most tested satellites ranging from 20-30 m square pixels, while
HJ2 has 6 km square pixels and WorldView-3 has highly sensitive 3.7 m square
pixels. Spectral resolution varies as well across the tested satellites, from 0.3 nm
for GHGSat-C and 200 nm for Sentinel-2 and LandSat 8/9 (Jacob et al., 2022),
discussed further in the SI, Section S2. See the SI, Section S2 for additional
discussion of the capabilities of each satellite system.

Table 1. Key characteristics of each participating satellite constellation, from lowest to highest swath width, which is roughly proportional to an instrument's
minimum methane detection limit. Global coverage refers to a configuration that passively covers most of Earth's surface over some number of orbits, while
targeted coverage refers to a "point-and-shoot" instrument that must be pointed to a particular location. Nadir pixel size is presented here. Constellation size
includes only active satellites. Accessing data from the GF5, ZY1, and HJ2 satellites requires permission from the Chinese government. Adapted with permission
from (Sherwin et al., 2023).

| Satellite | Coverage | Constellation size | Swath [km] | Pixel size [m] | ~Revisit time (per satellite) | Data availability | Source |
|---|---|---|---|---|---|---|---|
| GHGSat-C | Targeted | 8[§] | 12 | 25x25 | 14 days | Commercial | (ESA, 2022a; Jervis et al., 2021) |
| WorldView-3 | Targeted | 1 | 13.1 | 3.7x3.7 | 1 day[‡] | Commercial | (ESA, 2022b) |
| PRISMA | Targeted | 1 | 30 | 30x30 | 7 days | Public | (OHBI, 2022; ESA, 2012) |
| EnMAP | Targeted | 1 | 30 | 30x30 | 4 days[†] | Public | (EnMAP, 2023) |
| Gaofen 5 (GF5) | Targeted | 1 | 60 | 30x30 | 5-8 days[*] | Government | (Liu et al., 2019; Zhang et al., 2022; Luo et al., 2023) |
| Ziyuan 1 (ZY1) | Targeted | 1 | 60 | 30x30 | 1-3 days[*] | Government | (Song et al., 2022) |
| Landsat 8/9 | Global | 2 | 185 | 30x30 | 16 days | Public | (USGS, 2022) |
| Sentinel-2 | Global | 2 | 290 | 20x20 | 10 days | Public | (ESA, 2021) |
| Huanjing 2 (HJ2) | Unknown | 2 | 800 | 6x6 km | ≤4 days[*] | Government | (Zhong et al., 2021) |

[§]Three of these GHGSat C satellites were launched after the conclusion of testing.
[‡]WorldView-3 requires a 4.5-day repetition cycle for best resolution within 20° off nadir.
[†]EnMAP requires a 27-day repetition cycle for best resolution within 30° off (Jacob et al., 2022).
[*]Revisit times for GF5, ZY1, and HJ2 are inferred, at least in part, from overpass schedules submitted by NJU."

Page 15, section "Qualitatively assessing detection performance in the field", first sentence "The smallest emission detected by each team gives a sense of the minimum detection capabilities of each instrument," => I think that saying this sentence without nuances is dangerous, especially for satellites that have only been able to observe one emission during the experiment. The values given for each satellite are indeed relatively consistent with the literature for each of them, but in some cases, this leads to contradictions and can cause misunderstandings. For example, at the instrument level, EnMAP and PRISMA are very similar (with slight differences described in Roger et al. https://eartharxiv.org/repository/view/5235/), but the indicative detection limit estimated here is double for EnMAP than for PRISMA. The same happens with GF5, which is also similar to EnMAP and PRISMA, but GF5 has better spectral resolution at the same spatial resolution conditions, so we would expect a better detection capability than the other two satellites (this reasoning is explained in Jacob et al., 2022 https://acp.copernicus.org/articles/22/9617/2022/acp-22-9617-2022.html). I suggest rephrasing the sentence saying that the range of different flux rate emissions detected in this experiment gives an indication of the capabilities and, in the case of satellites not able to see the smallest emissions, of the limitations of each of the instruments. However, to set a detection limit for each of them, a larger number of detections is needed, ranging from true positives (when the satellite can see the emission) to false negatives (when the emission exists but the satellite cannot see it).

**Authors:** We have expanded our discussion of the detection capabilities of these instruments, combining our empirical findings for the controlled releases we conducted with other existing theoretical and simulation literature to present the most comprehensive picture we can of the likely detection capabilities of each system tested in this study.

**Manuscript, L469:** "The smallest emission detected by each team gives a rough upper bound on the lower detection capabilities of each instrument, at least in a desert environment with a known release location. We compare these smallest detected emissions with previous estimates of lower detection capabilities of each satellite. The smallest emission detected was 0.0332 [0.0328, 0.0336] t/h, identified by Maxar using WorldView-3, shown in **Error! Reference source not found.**. Kayrros also detected an emission below 0.1 t/h using WorldView-3. This is consistent with previous estimates of lower detection capabilities, with Sánchez-García et al. detecting an emission estimated at ~0.040 t/hr in Turkmenistan using WorldView-3 (Sánchez-García et al., 2022).

Orbio Earth, Maxar, and GHGSat all detected a 1.19 [1.15, 1.23] t/h emission using Sentinel-2, with errors ranging from -8% to +170%. Orbio Earth detected a

1.05 [0.99, 1.10] t/h emission to within ±47%. These emissions are 15-25% below the smallest emission detected using Sentinel-2 in any previous satellite controlled methane release test, and consistent with simulation-based estimates (Sherwin et al., 2023; Gorroño et al., 2023). The story is similar for LandSat 8/9, with the smallest detected emission at 1.39 [1.34, 1.43] t/h. This is also slightly below estimated lower detection capabilities in the literature (Jacob et al., 2022).

The smallest emission detected via PRISMA was 0.414 [0.410, 0.417] t/h smaller than the 0.5-2.0 t/h estimated by Guanter et al. as PRISMA's lower detection threshold (Guanter et al., 2021). The smallest detected emissions for the remaining satellites are 1.10 [1.06, 1.13] t/h for EnMAP, 1.26 [0.26, 2.26] t/h for GF5, and 1.03 [0.98, 1.09] t/h for ZY1. However, given that the technical characteristics of these three satellites are similar to PRISMA, they can likely be used to detect emissions below 1 t/h, at least under favorable environmental conditions (Jacob et al., 2022; Roger et al., 2023).

GHGSat correctly detected and quantified the only nonzero release for which GHGSat-C collected data and passed quality control, which was 0.401 [0.399, 0.404] t/h, roughly double the smallest release GHGSat quantified using the same satellite system in (Sherwin et al., 2023). GHGSat's lower detection threshold is estimated at 0.1-0.2 t/h (Jacob et al., 2022). HJ2B was not tasked during any active releases, meaning that future testing is needed to assess its detection capabilities."

Discussion, beginning of the second paragraph: I would say that the high detection limit of LanSat and Sentinel-2 is more related to their low spectral resolution (bandwidth) than to the swath. WV3 also has a relatively low spectral resolution compared to hyperspectral satellites (EnMAP, PRISMA, GF5, ZY1, HJ2B, and also GHGSat), but this is compensated by its very high spatial resolution. Indeed, spectral resolution is an essential parameter in methane detection capability (Jacob et al., 2022) that is not considered in this paper.

**Authors:** We agree that spectral resolution is very important for methane remote sensing. We now discuss it in more detail as follows,

**Manuscript, L659:** "Detection limits appear to improve with smaller swath width and pixel size, and with higher spectral resolution. Global-coverage satellites such as LandSat 8/9 and Sentinel-2, with swaths of 185 and 290 km, respectively, and spectral resolution 20-650 times coarser than the hyperspectral instruments (EnMAP, PRISMA, GF5, ZY1, HJ2B, and GHGSat), have higher detection limits. See the SI, Section S2 for additional discussion of spectral resolution. Our results are consistent with (Gorroño et al., 2023), whose simulation-based approach suggests that such instruments have a best-case minimum detection limit of roughly 1 t/h. Targeted satellites with swaths of 30-60 km, including EnMAP, GF5, PRISMA, and ZY1 (EnMAP, 2023; Liu et al., 2019; OHBI, 2022; Song et al., 2022), all reliably saw emissions of ~1 t/h. Of these, only PRISMA has had the opportunity to be tested with emission fluxes below 1 t/h, correctly detecting 0.413 [0.410, 0.417] t/h, the smallest emission given to PRISMA. GHGSat correctly detected 0.401 [0.399, 0.403] t/h, with quantification accuracy within ±20%, using their GHGSat-C-series satellite, with a swath width of 12 km. Estimates for smaller emission sizes were filtered due to clouds, but in previous testing GHGSat successfully detected an 0.197 [0.187, 0.208] t/h emission and quantified it with similar accuracy, suggesting that the system may be capable of seeing emissions even smaller than 200 kg/h.

Maxar successfully detected emissions as low as 0.0332 [0.0328, 0.0336] t/h using the WorldView-3 satellite, with swath width 13.1 km. Two teams successfully detected emissions below 0.1 t/h using WorldView-3, while two teams applied more conservative criteria and detected only emissions above 0.5 t/h. Although Maxar has a coarser spectral resolution than hyperspectral instruments, its very high spatial resolution enables heightened sensitivity."

Discussion, second paragraph, sentence "Of these, only PRISMA was given smaller emissions, with three of four teams correctly detecting 0.413 [0.410, 0.417] t/h, the smallest emission given to PRISMA. ": Again, I think that saying this sentence as it is is dangerous because it can be easily misinterpreted, implying that PRISMA has the best detection capability among the four hyperspectral satellites when EnMAP, GF5, and ZY1 have only had one detection occasion and have not had the opportunity to test their ability with smaller fluxes. I proposed to change this sentence to "Of these, only PRISMA has had the opportunity to be tested with emission fluxes below 1 t/h, correctly detecting 0.413 [0.410, 0.417] t/h, the smallest emission given to PRISMA".

**Authors:** We now used the proposed language for this sentence, in L667.

Considering that one of the major elements in the manuscript is methane (the second most important greenhouse gas whose anthropogenic emissions should be avoided due to its impact on global warming), for transparency, I would appreciate a section where authors clarify the total amount of methane released during the experiment. This can be addressed with a simple sentence in, for example, the experimental design section or with a separate section in the SI. For clarity, it would also be useful to compare that total amount emitted to a well-documented emissions event (equals x% of what was emitted in said event) or estimate for a region or sector to put readers in perspective.

**Authors:** We have added a section to the SI outlining total emissions from the satellite portion of our 2022 testing (we also tested other technologies during the same two-month period). We highlight in the manuscript that satellites regularly detect individual sources that emit in one hour over five times the total emissions required to test these satellites over two months.

**Manuscript, L770:** "It is important to note that conducting this test did require the release of considerable amounts of methane into the atmosphere. We estimate total emissions from the satellite testing discussed in this paper at 7.7 t(CH$_4$)/h, discussed further in the SI, Section S1.5. However, this pales in comparison with anthropogenic emissions occurring across the globe. Lauvaux et al. identify over 1000 emission sources across the world emitting at least 7.7 t(CH$_4$) every hour, in some cases over 50 times as much every hour (Lauvaux et al., 2022). If this work assists in accelerating mitigation of even one of these emissions by even a single hour, e.g. by ensuring key decision-makers view satellite-based methane detection and quantification as reliable, we will have broken even from a methane emissions perspective."

**SI, Section S.1.5:** "**Total emissions during testing**
We estimate total methane emissions released while testing satellites, not including methane released during the two-month testing period to evaluate other technologies. As a rule, we held each release at a constant volume for 15 minutes before a satellite passed overhead, and for five minutes afterward. For simplicity, we assume by default that all releases were held for 20 minutes at the 5-minute average volume before the satellite passed overhead. Across the full dataset, there are 47 unique satellite overpass timestamps during nonzero methane releases. This naïve approach, which does not account for near-simultaneous overpasses, would estimate total emissions at 10.7 t(CH$_4$).

In some cases, multiple satellites passed overhead within 20 minutes or less. In those cases, we subtract out methane associated with any overlapping period. This occurred on October 10th, 17th, 26th, and 29th, as well as November 8th, 10th, 15th, 17th, 18th, 28th, and 29th. See the replication code in GitHub, in the script entitled "SatelliteTestingMain.ipynb."

After accounting for these overlapping release periods for satellite overpasses occurring close in time, total estimated emissions from this test fall to 7.7 t(CH$_4$)."

Section S.4.6.1: I think that adding the wind speed data from the reanalysis product that each group used for the initial estimate indicated in each image would help a lot in the interpretation of the results.

**Authors:** We have added the 5-minute average wind speed and direction, as measured from our in situ 10-m anemometer for each of these images. We feel in situ measurements are more useful than the reanalysis estimates, as they are a closer reflection of what was happening on the ground (it is not uncommon for reanalysis data to point in the wrong direction).

Minor comments:

Table 1: Coverage HJ2. In the paragraph just before the table, the authors say that it is not clear whether HJ2 is targeted or global, but in the table, it is classified as global. If the text is correct, perhaps the table should read "no data" or similar?

> **Authors:** We have updated this text to read "Unknown", although the satellite's characteristics suggest it is likely closer to global coverage.

In Table 1, the revisit time that the authors indicate for WV3 and EnMAP is actually the repetition cycle. For PRISMA, they provide the revisit time but do not specify the repetition cycle. For consistency, I suggest indicating in the table the revisit time (WV3=1 day, PRISMA=7 days, and EnMAP=4 days) and in the annotations the repetition cycle (best resolution by looking at nadir). For PRISMA, the repetition cycle is 29 days https://www.eoportal.org/satellite-missions/prisma-hyperspectral#launch

> **Authors:** We have updated Table 1 as recommended, including revisit time in the table itself, with annotations for repetition cycle for best resolution:

> > **Manuscript, Table 1:**

"Table 2. Key characteristics of each participating satellite constellation, from lowest to highest swath width, which is roughly proportional to an instrument's
minimum methane detection limit. Global coverage refers to a configuration that passively covers most of Earth's surface over some number of orbits, while
targeted coverage refers to a "point-and-shoot" instrument that must be pointed to a particular location. Nadir pixel size is presented here. Constellation size
includes only active satellites. Accessing data from the GF5, ZY1, and HJ2 satellites requires permission from the Chinese government. Adapted with permission
from (Sherwin et al., 2023).

| Satellite | Coverage | Constellation size | Swath [km] | Pixel size [m] | ~Revisit time (per satellite) | Data availability | Source |
|---|---|---|---|---|---|---|---|
| GHGSat-C | Targeted | 8[§] | 12 | 25x25 | 14 days | Commercial | (ESA, 2022a; Jervis et al., 2021) |
| WorldView-3 | Targeted | 1 | 13.1 | 3.7x3.7 | 1 day[‡] | Commercial | (ESA, 2022b) |
| PRISMA | Targeted | 1 | 30 | 30x30 | 7 days | Public | (OHBI, 2022; ESA, 2012) |
| EnMAP | Targeted | 1 | 30 | 30x30 | 4 days[†] | Public | (EnMAP, 2023) |
| Gaofen 5 (GF5) | Targeted | 1 | 60 | 30x30 | 5-8 days[*] | Government | (Liu et al., 2019; Zhang et al., 2022; Luo et al., 2023) |
| Ziyuan 1 (ZY1) | Targeted | 1 | 60 | 30x30 | 1-3 days[*] | Government | (Song et al., 2022) |
| Landsat 8/9 | Global | 2 | 185 | 30x30 | 16 days | Public | (USGS, 2022) |
| Sentinel-2 | Global | 2 | 290 | 20x20 | 10 days | Public | (ESA, 2021) |
| Huanjing 2 (HJ2) | Unknown | 2 | 800 | 6x6 km | ≤4 days[*] | Government | (Zhong et al., 2021) |

[§]Three of these GHGSat C satellites were launched after the conclusion of testing.
[‡]WorldView-3 requires a 4.5-day repetition cycle for best resolution within 20° off nadir.
[†]EnMAP requires a 27-day repetition cycle for best resolution within 30° off (Jacob et al., 2022).
[*]Revisit times for GF5, ZY1, and HJ2 are inferred, at least in part, from overpass schedules submitted by NJU."

Page 7, last paragraph, when the authors say "or the precise location of ground-based equipment.", I would suggest, for clarity, adding "within the provided location coordinates" or similar as, in the first paragraph of the section, the authors are saying that "Participating teams were aware of the precise location coordinates of the test".

> **Authors:** To clarify what information participating teams did and did not have at the time of testing, we have updated this section as follows:
>
> > **Manuscript, L151:** "The Stanford ground team and contract personnel operating equipment communicated no information to participating teams regarding metered flow rates or metered wind speed or direction. Participating teams were aware of the precise location coordinates of the test, but were not informed of the precise configuration of ground-based equipment within the test site."

Section "First-time single-blind detections from Chinese and European satellites" I suggest changing the title to "First-time single-blind detections from some of the satellites" or similar, as it may suggest that it is the first single-blind detection test from all European satellites taking part.

> **Authors:** This section header now reads "**First-time single-blind detections from three satellites**". The three satellites in question are EnMAP, ZY1, and GF5. Sadly, the fact that the single HJ2B acquisition was rescheduled without notice to a time at which we were not releasing methane means that there was not an opportunity for this fourth satellite to have its first single-blind methane detections.

Page 10 section "First-time single-blind detections from Chinese and European satellites" end of the paragraph: EnMAP has also been used and evaluated for methane detection in Roger et al. 2023 (still in preprint) https://eartharxiv.org/repository/view/5235/ which I think should be taken into account in the references.

> **Authors:** We now cite this preprint, which compares EnMAP retrievals with PRISMA retrievals, but does not have metered ground-truth emission rates, as we do.
>
> > **Manuscript, L234:** "Roger et al. compare EnMAP retrievals with the single-blind-validated PRISMA satellite as a benchmark, finding promising results, especially for offshore emissions of 1 t/h or more (Roger et al., 2023)."

Figure 3: EnMAP/NJU window => I think that for consistency, it makes more sense to show the background Google Earth map with nothing overlaid since the authors already show the retrieval of the image "with nothing" in section 4 of the SI along with the rest of the retrievals, although this is not nothing critical.

**Authors:** As requested, we have updated Figure 3 to show the Google Earth background map instead of the unmasked retrieval.

In the figure caption of Figure 3, it is not mentioned what the * of the 1.3 t/h of Gaofen5 is

**Authors:** We now clarify the meaning of this asterisk in Figure 3 as follows,

> **Manuscript, L259:** "*The Gaofen 5 measurement was rescheduled without notice to a time that happened to be one minute after releases had concluded for a different satellite, resulting in artificially high variability in the metered ground-truth flow rate."

In Figure 3, Gaofen 5 and Ziyuan 1 should go without a hyphen (-) for consistency with the rest of the text. Similarly, both satellites are presented as Gaofen 5 and Gaofen5-02 and Ziyuan 1 and Ziyuan 1-02 inconsistently throughout the text.

**Authors:** We have removed extra hyphens, and now refer consistently to these satellites as Ziyuan 1 and Gaofen 5 throughout the text. We do clarify in the main text and in the SI which edition of these satellite series were tested in this work.

> **Manuscript, L77:** "In addition, several methane-sensing satellites have launched since the previous test concluded in 2021, including the German EnMAP system and the 02 edition of the Chinese Gaofen 5 Advanced Hyperspectral Imager (GF5) and the 02E edition of the Ziyuan 1 Advanced Hyperspectral Imager (ZY1) (EnMAP, 2023; Xinhua, 2022; Song et al., 2022)."

Bibliographic references should be corrected and adapted to a single format. Some of the references are listed twice in the bibliography, others are not updated, and many have errors:

- References 2 and 44 are the same, but 44 is not updated, referring to the preprint of the paper.
- Reference 4: the correct link is this: https://amt.copernicus.org/articles/15/1657/2022/amt-15-1657-2022.html (no longer in discussion)
- References 15 and 51 are the same.
- The link in reference 57 does not work, but I would say it is the same as in reference 21
- References 32 and 35 are the same.
- Reference 45 is not updated. The revised and published paper is this: https://www.sciencedirect.com/science/article/abs/pii/S0034425721003916
- Reference 70, the link does not work.

**Authors:** We have updated these references, removed duplicates, and fixed broken/outdated links.

Section S2. Participating satellites: in the description of all satellites (except ZY1), the spectral resolutions (Bandwidth) and spatial resolutions (pixel size) are missing, which are important parameters that significantly determine the sensitivity of the satellite to methane.

**Authors:** We have added spectral and spatial resolution information to all satellite descriptions in the SI, Section S2, and now refer to this section in the discussion:

**Manuscript, L660:** "Global-coverage satellites such as LandSat 8/9 and Sentinel-2, with swaths of 185 and 290 km, respectively, and spectral resolution 20-650 times coarser than the hyperspectral instruments (EnMAP, PRISMA, GF5, ZY1, HJ2B, and GHGSat), have higher detection limits. See the SI, Section S2 for additional discussion of spectral resolution."

Section S.2.6. PRISMA: "operating with a 7-day maximum revisit frequency." => operating with a 7-day maximum revisit frequency and 29-day nadir revisit frequency.

**Authors:** We have adopted the suggested text.

Section S.4.6.1: I assume that the value of the estimated flux for each group in each of the images (in white in the figure with the masked plume) corresponds to stage 1, which is why the Maxar PRISMA estimates are not shown. If so (or not), I think it should be indicated at the beginning of the section or in the figure captions, and also the reason why the Maxar PRISMA data is missing.

**Authors:** We now include Maxar's Stage 1 emission rate inset in these images, with a note in this figure caption as follows:

**SI, Section S4.6.1, Figure 41:** "For nonzero estimated emissions, mean estimated emission rate in white inset text (in this case, the Stage 1 estimate submitted past the deadline by Maxar)."

Page 6, last paragraph, and page 23, last paragraph, th and nd to November 15 and November 22, are missing (for consistency with the rest of the dates).

**Authors:** Fixed

**References**

[revised manuscript text omitted]